# On the Generalization of Models Trained with SGD: Information-Theoretic Bounds and Implications

**Ziqiao Wang**
University of Ottawa
zwang286@uottawa.ca

**Yongyi Mao**
University of Ottawa
ymao@uottawa.ca

## Abstract

This paper follows up on a recent work of Neu et al. (2021) and presents some new information-theoretic upper bounds for the generalization error of machine learning models, such as neural networks, trained with SGD. We apply these bounds to analyzing the generalization behaviour of linear and two-layer ReLU networks. Experimental study of these bounds provide some insights on the SGD training of neural networks. They also point to a new and simple regularization scheme which we show performs comparably to the current state of the art.

## 1 Introduction

The observation that high capacity deep neural networks trained with mini-batched stochastic gradient descent, referred to SGD in this paper, tend to generalize well (Zhang et al., 2017) contradicts the classical wisdom in statistical learning theory (e.g., Vapnik (1998) ) and has stimulated intense research interest in understanding the generalization behaviour of modern neural networks.

In this direction, generalization bounds for over-parameterized neural networks are obtained (Allen-Zhu et al., 2019; Bartlett et al., 2017; Neyshabur et al., 2015; 2018a;b; Arora et al., 2018; 2019) and a curious "double descent" phenomenon is observed and analyzed (Belkin et al., 2019; Nakkiran et al., 2019; Yang et al., 2020). Built on a connection between stability and generalization (Bousquet & Elisseeff, 2002), a stability-based bound is first presented in Hardt et al. (2016), followed by a surge of research effort exploiting similar approaches (London, 2017; Chen et al., 2018; Feldman & Vondrak, 2019; Lei & Ying, 2020; Bassily et al., 2020). Information-theoretic bounding techniques established recently (Russo & Zou, 2016; 2019; Xu & Raginsky, 2017; Asadi et al., 2018; Bu et al., 2020; Steinke & Zakynthinou, 2020; Dwork et al., 2015; Bassily et al., 2018; Asadi & Abbe, 2020; Hafez-Kolahi et al., 2020; Zhou et al., 2020) have also demonstrated great power in analyzing SGD-like algorithms. For example, Pensia et al. (2018) is the first to utilize information-theoretic bound in analyzing the generalization ability of SGLD (Gelfand & Mitter, 1991; Welling & Teh, 2011). The bound was subsquently improved by Negrea et al. (2019); Haghifam et al. (2020); Rodríguez-Gálvez et al. (2020); Wang et al. (2021b). Inspired by the work of Pensia et al. (2018), Neu et al. (2021) presents an information-theoretic analysis of the models trained with SGD. The analysis of Neu et al. (2021) constructs an auxiliary weight process parallel to SGD training and upper-bounds the generalization error through this auxiliary process.

Another line of research connects the generalization of neural networks with the flatness of loss minima (Hochreiter & Schmidhuber, 1997) found by SGD or its variant (Keskar et al., 2017; Dinh et al., 2017; Dziugaite & Roy, 2017; Neyshabur et al., 2017; Chaudhari et al., 2017; Jastrzebski et al., 2017; Jiang et al., 2019; Zheng et al., 2021; Foret et al., 2020). This understanding has led to the discovery of new SGD-based training algorithms for improved generalization. For example, in a concurrent development by Zheng et al. (2021) and Foret et al. (2020), a local "max-pooling" operation is applied to the loss landscape prior to the SGD updates. This approach, referred to as AMP (Zheng et al., 2021) or SAM (Foret et al., 2020), is shown to make SGD favor flatter minima and achieve the state-of-the-art performance among various competitive regularization schemes.

In this paper, we focus on investigating the generalization of machine learning models trained with SGD. Although we are primarily motivated by the curiosity to understanding neural networks, the results of this paper in fact apply broadly to any model trained with SGD.

This work follows the same construction of the auxiliary weight process in Neu et al. (2021) and develops upper bounds of generalization error that extend the work of Neu et al. (2021). Like those in Neu et al. (2021), the bounds we obtain can be decomposed into two terms, one measuring the impact of training trajectories ("the trajectory term") and the other measuring the impact of the flatness of the found solution ("the flatness term[1]"). Having an identical flatness term as that in Neu et al. (2021), empirical evidence hints that our bounds have a improved trajectory term. Figure 1 shows an experimental comparison of the trajectory term in our bound (Theorem 2) with that in the bound in Neu et al. (2021) (re-stated as Lemma 2 in this paper) for two neural network models. The trajectory terms are compared in a deterministic setting for two different values of $\sigma$ (the variance parameter of the noise in the auxiliary weight process, details given in appropriate context and in Appendix E.2).

The trajectory term in the bounds of Neu et al. (2021) accumulates two terms over training steps: *local gradient sensitivity*, measuring the sensitivity of the SGD gradient signal to weight perturbations, and *gradient dispersion*[2], measuring the extent to which the gradient signal spreads around its mean. Usually gradient dispersion vanishes with training iterations but local gradient sensitivity does not. Our improvement over Neu et al. (2021) is achieved by removing the local sensitivity term and involving the logarithmic of a revised gradient dispersion. Although our new definition of gradient dispersion is in general larger than that in Neu et al. (2021), as long as the number of iterations is not too small, the fact that our gradient dispersion also vanishes with training allows our bounds to be tighter than Neu et al. (2021).

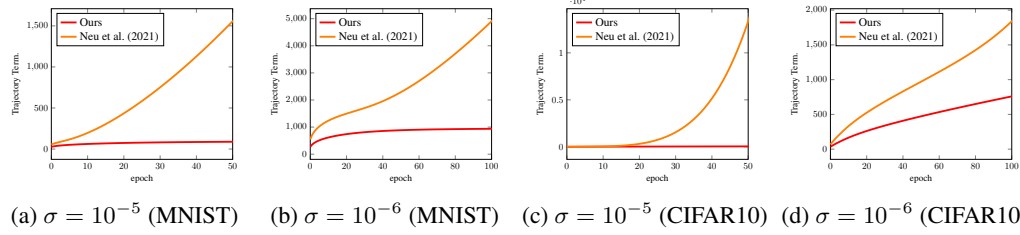

(a) $\sigma = 10^{-5}$ (MNIST)    (b) $\sigma = 10^{-6}$ (MNIST)    (c) $\sigma = 10^{-5}$ (CIFAR10)    (d) $\sigma = 10^{-6}$ (CIFAR10)

Figure 1: Comparison of the trajectory term between our bound (Theorem 2) and the bound in Neu et al. (2021). (a)(b) MLP trained on MNIST. (c)(d) CNN trained on CIFAR-10.

We also provide an application of our bounds in analyzing the generalization behaviour of linear and two-layer ReLU networks, where we show that the activation state in ReLU networks plays an important role in generalization.

It is remarkable that removing the local gradient sensitivity term makes our bound have a simple closed form (after optimizing the noise variances), much easier to evaluate. We empirically validate the derived bounds, and provide various insights pertaining to the generalization behavior of models trained with SGD. For example, gradient dispersion is seen to reveal a double descent phenomenon with respect to training epochs, where the valley in the double descent curve appears to mark the great divide between the "generalization regime" and the "memorization regime" of training. Furthering from this observation, we also show that it is possible to reduce the memorization effect by dynamically clipping the gradient and reducing its dispersion.

Our bounds also inspire a natural and simple solution to alleviate generalization error. Specifically, we propose a new training scheme, referred to as *Gaussian model perturbation* (GMP), aiming at reducing the flatness term of the bounds. This scheme effectively applies a local "average pooling" to the empirical risk surface prior to SGD, greatly resembling the "max-pooling" approach adopted in AMP (Zheng et al., 2021). We demonstrate experimentally that GMP achieves a competitive performance with the current art of regularization schemes.

---

[1]We note this term is correlated with flatness rather than precisely measures the flatness.

[2]The quantity is often referred to as gradient variance in the literature, but we prefer "dispersion" to "variance" so as to better comply with the mathematical conventions and avoid possible confusion.

Proofs, additional discussions and experimental results are presented in Appendices.

**Other Related Literature** Gradient dispersion is mostly studied from optimization perspectives (Bottou et al., 2018; Roux et al., 2012; Johnson & Zhang, 2013; Wen et al., 2020; Faghri et al., 2020). Prior to this work, only a few works relate gradient dispersion with the generalization behaviour of the networks. In Neu et al. (2021) and Wang et al. (2021b), gradient dispersion also appears in the generalization bounds. and there are limited studies characterizing the generalization performance and the gradient variance. Particularly, in Jiang et al. (2019), gradient dispersion is argued to capture a notion of "flatness" of the local minima of the loss landscape, thereby correlating with generalization. Injecting noise in the training process has been proposed in various regularization schemes (Bishop, 1995; Camuto et al., 2020; 2021; Srivastava et al., 2014; Wei et al., 2020). But unlike GMP derived in this paper, where noise is injected to the model parameters, noise in those schemes is injected either to the training data or to the network activation. In addition, noise can also be implicitly injected during training, for example, by adjusting the momentum terms Xie et al. (2021b). The training objective in Xie et al. (2021a) is similar to our GMP but their NVRM-SGD minimizes a region directly while our GMP considers a trade-off between the empirical loss and the flatness term. Gradient clipping is a common technique for preventing gradient exploding (see, e.g., Merity et al. (2018); Peters et al. (2018)). This technique is also used in Zhang et al. (2019) to accelerate training. In this paper, gradient clipping is used to investigate and control the impact of gradient dispersion on generalization error.

## 2 PRELIMINARIES

**Population Risk, Empirical Risk and Generalization Error** Unless otherwise noted, a random variable will be denoted by a capitalized letter (e.g., $Z$), and its realization denoted by the corresponding lower-case letter (e.g. $z$). Let $\mathcal{Z}$ be the instance space of interest and $\mu$ be an unknown distribution on $\mathcal{Z}$, specifying random variable $Z$. Let $\mathcal{W} \subseteq \mathbb{R}^d$ be the space of hypotheses. Suppose that a training sample $S = (Z_1, Z_2, \ldots, Z_n)$ is drawn i.i.d. from $\mu$ and that a stochastic learning algorithm $\mathcal{A}$ takes $S$ as its input and outputs a hypothesis $W \in \mathcal{W}$ according to some conditional distribution $P_{W|S}$ mapping $\mathcal{Z}^n$ to $\mathcal{W}$. Let $\ell : \mathcal{W} \times \mathcal{Z} \to \mathbb{R}^+$ be a loss function, where $\ell(w, z)$ measures the "unfitness" or "error" of any $z \in \mathcal{Z}$ with respect to a hypothesis $w \in \mathcal{W}$. The population risk, for any $w \in \mathcal{W}$, is defined as

$$L_\mu(w) \triangleq \mathbb{E}_{Z \sim \mu}[\ell(w, Z)].$$

The goal of learning is to find a hypothesis $w$ that minimizes the population risk. But since $\mu$ is only partially accessible via the sample $S$, in practice, we instead turn to the empirical risk, defined as

$$L_S(w) \triangleq \frac{1}{n} \sum_{i=1}^{n} \ell(w, Z_i).$$

The expected generalization error of the learning algorithm $\mathcal{A}$ is then defined as

$$\text{gen}(\mu, P_{W|S}) \triangleq \mathbb{E}_{W,S}[L_\mu(W) - L_S(W)],$$

where the expectation is taken over the joint distribution of $(S, W)$ (i.e., $\mu^n \otimes P_{W|S}$).

Throughout this paper, we take $\ell$ as a continuous function (adopting the usual notion "surrogate loss" (Shalev-Shwartz & Ben-David, 2014) ). Additionally, we assume that $\ell$ is differentiable almost everywhere with respect to both $w$ and $z$. Furthermore we assume that $\ell(w, Z)$ is $R$-subgaussian[3] for any $w \in \mathcal{W}$. Note that a bounded loss is guaranteed to be subgaussian. Let $I(X; Y)$ denote the mutual information (Cover & Thomas, 2012) between any pair of random variables $(X, Y)$. The following result is known.

**Lemma 1** (Xu & Raginsky (2017, Theorem 1.)). *Assume the loss $\ell(w, Z)$ is $R$-subgaussian for any $w \in \mathcal{W}$. The generalization error of $\mathcal{A}$ is bounded by*

$$|\text{gen}(\mu, P_{W|S})| \leq \sqrt{\frac{2R^2}{n} I(W; S)},$$

---

[3]A random variable $X$ is $R$-subgaussian if for any $\rho$, $\log \mathbb{E} \exp \left( \rho \left( X - \mathbb{E}X \right) \right) \leq \rho^2 R^2 / 2$.

**Stochastic Gradient Descent**    We now restrict the learning algorithm $\mathcal{A}$ to be the mini-batched SGD algorithm for empirical risk minimization. For each training epoch, the dataset $S$ is randomly split into $m$ disjoint mini-batches, each having size $b$, namely, $n = mb$. Based on each batch, one parameter update is performed. Specifically, let $B_t$ denote the batch used for the $t^{\text{th}}$ update. Define

$$g(w, B_t) \triangleq \frac{1}{b} \sum_{z \in B_t} \nabla_w \ell(w, z),$$

namely, $g(w, B_t)$ is the average gradient computed for the batch $B_t$ with respect to parameter $w$. The rule for the $t^{\text{th}}$ parameter update is then

$$W_t \triangleq W_{t-1} - \lambda_t g(W_{t-1}, B_t),$$

where $\lambda_t$ is the learning rate at the step $t$. The initial parameter setting $W_0$ is assumed to be drawn from the zero-mean spherical Gaussian $\mathcal{N}(0, \sigma_0^2 \mathbf{I}_d)$ with variance $\sigma_0^2$ in each dimension. We will assume that the SGD algorithm stops after $T$ updates and outputs $W_T$ as the learned model parameter.

Given the training sample $S$, let $\xi$ govern the randomness in the sequence $(B_1, B_2, \ldots, B_T)$ of batches. For the simplicity of notion, we will fix the configuration of $\xi$. That is, we will assume a fixed "batching trajectory", or a fixed way to shuffle the example indices $\{1, \ldots, n\}$ and divide them into $m$ batches in each epoch. The presented generalization bounds of this paper can be extended to the case where the batching trajectory is uniformly random (as we set up above). This merely involves averaging over all batching trajectories or taking expectation over $\xi$.

**Auxiliary Weight Process**    We now associate with the SGD algorithm an auxiliary weight process $\{\widetilde{W}_t\}$. Let $\sigma_1, \sigma_2, \ldots, \sigma_T$ be a sequence of positive real numbers. Define

$$\widetilde{W}_0 \triangleq W_0, \quad \text{and} \quad \widetilde{W}_t \triangleq \widetilde{W}_{t-1} - \lambda_t g(W_{t-1}, B_t) + N_t, \text{ for } t > 0,$$

where $N_t \sim \mathcal{N}(0, \sigma_t^2 \mathbf{I}_d)$ is a Gaussian noise. The relationship between this auxiliary weight process $\{\widetilde{W}_t\}$ and the weight process $\{W_t\}$ in SGD is shown in the Bayesian network below.

$$
\begin{array}{ccccccccccc}
& & N_1 & & N_2 & & \cdots & & N_{T-1} & & N_T \\
& & \downarrow & & \downarrow & & & & \downarrow & & \downarrow \\
\widetilde{W}_0 & \to & \widetilde{W}_1 & \to & \widetilde{W}_2 & \to & \cdots & \to & \widetilde{W}_{T-1} & \to & \widetilde{W}_T \\
\| & \nearrow & & \nearrow & & \nearrow & & & & \nearrow & \\
W_0 & \to & W_1 & \to & W_2 & \to & \cdots & \to & W_{T-1} & \to & W_T
\end{array}
$$

Let $\Delta_t = \sum_{\tau=1}^{t} N_\tau$. Noting that the weight updates in $\{\widetilde{W}_t\}$ uses the same gradient signal as that used in $\{W_t\}$ (which depends on $W_{t-1}$ not $\widetilde{W}_{t-1}$), it is immediate that $\widetilde{W}_t = W_t + \Delta_t$. Note that this auxiliary process follows the same construction as Neu et al. (2021), which we will use to study the generalization error of SGD. To that end, let's define *gradient dispersion* by

$$\mathbb{V}_t(w) \triangleq \mathbb{E}\left[ ||g(w, B_t) - \mathbb{E}\left[\nabla_w \ell(W, Z)\right]||_2^2 \right], \tag{1}$$

where the inside expectation is taken over $(W, Z) \sim \mu \otimes P_{W|Z}$. For a given sample $s \in \mathcal{Z}^n$, define

$$\gamma(w, s) \triangleq \mathbb{E}\left[ L_s(w + \Delta_T) - L_s(w) \right],$$

where the expectation is taken over $\Delta_T$ and $L_s(w)$ is the empirical risk of $s$ at parameter $w$.

In the remainder of the paper, let $S'$ denote another sample drawn from $\mu^n$, independent of all other random variables. The main generalization bound in Neu et al. (2021) is re-stated below.

**Lemma 2** ((Neu et al., 2021, Theorem 1.)). *The generalization error of SGD is upper bounded by*

$$|\text{gen}(\mu, P_{W_T|S})| \le 2\sqrt{\frac{2R^2}{n} \sum_{t=1}^{T} \frac{\lambda_t^2}{\sigma_t^2} \mathbb{E}\left[ \Psi(W_{t-1}) + \widetilde{\mathbb{V}}_t(W_{t-1}) \right]} + |\mathbb{E}\left[ \gamma(W_T, S) - \gamma(W_T, S') \right]|,$$

*where $\Psi(w_{t-1}) \triangleq \mathbb{E}\left[ ||\mathbb{E}\left[\nabla_w \ell(w_{t-1}, Z)\right] - \mathbb{E}\left[\nabla_w \ell(w_{t-1} + \zeta, Z)\right]||_2^2 \right]$, $\zeta \sim \mathcal{N}(0, \sum_{i=1}^{t-1} \sigma_i^2 \mathbf{I}_d)$ and $\widetilde{\mathbb{V}}_t(w) \triangleq \mathbb{E}\left[ ||g(w, B_t) - \mathbb{E}\left[\nabla_w \ell(w, Z)\right]||_2^2 \right].$*

The term $\Psi(w_{t-1})$ in the bound is referred to as "local gradient sensitivity" in Neu et al. (2021). Note that the inside expectation of $\widetilde{\mathbb{V}}_t(w)$ is taken over $Z \sim \mu$ instead of $(W, Z) \sim \mu \otimes P_{W|Z}$. Thus, $\widetilde{\mathbb{V}}_t(w)$ is in general not worse than our gradient dispersion $\mathbb{V}_t(w)$ for a fixed $w$ (see Figure 5 (d) in Appendix E.2). However, the difference between these two terms is very small when $W$ is close to local minima due to the tiny gradient norm. In addition, the definition of our gradient dispersion is implicitly used in a bound in Section 5.2 of Neu et al. (2021), which can be regarded as a weaker version of Eq. 2 in our Theorem 2.

## 3  NEW GENERALIZATION BOUNDS FOR SGD

Removal of the local sensitivity term $\Psi(w_{t-1})$ requires invoking a special instance of the HWI inequality (Raginsky & Sason, 2018, Lemma 3.4.2), which we first state.

**Lemma 3.** *Let $X$ and $Y$ be two random vectors in $\mathbb{R}^d$, and let $N \sim \mathcal{N}(0, \mathbf{I}_d)$ be independent of $(X, Y)$. Then, for any $t, t' > 0$, $\mathrm{D}_{\mathrm{KL}}(P_{X+\sqrt{t}N} || P_{Y+\sqrt{t'}N}) \leq \frac{1}{2t'}\mathbb{E}\left[||X - Y||_2^2\right] + \frac{d}{2}(\log\frac{t'}{t} + \frac{t}{t'} - 1)$, where $\mathrm{D}_{\mathrm{KL}}$ is the KL divergence.*

Note that Neu et al. (2021) also makes uses of a similar lemma in their revision. However, the development in Neu et al. (2021) requires constructing a ghost auxiliary weight process in which an independent noise perturbation $\Delta_t'$ is introduced. Due to the mismatch between $\Delta_t$ and $\Delta_t'$, the local gradient sensitivity term $\Psi(W_{t-1})$ appears in their bound. In this paper, we waive this $\Delta_t'$ by using the following lemma.

**Lemma 4.** *Let random variables $X, Y$ and $\Delta$ be independent of $N \sim \mathcal{N}(0, \mathbf{I}_d)$. Then for any $\sigma > 0$, any $\mathbb{R}^d$-valued function $f$, and any random variable $\Omega \in \mathbb{R}^d$ that is a function of $Y$, we have*

$$I(f(Y + \Delta, X) + \sigma N; X|Y) \leq \frac{d}{2}\mathbb{E}\left[\log\left(\frac{\mathbb{E}\left[||f(Y + \Delta, X) - \Omega||^2\right]}{d\sigma^2} + 1\right)\right].$$

Note that the outside expectation is taken over $Y$ and the inside expectation is taken over $(X, \Delta)$. Then, exploiting Lemma 4 by letting $X = S$, $Y = \widetilde{W}$ and $\Omega = \mathbb{E}\left[g(\widetilde{w} - \Delta, Z)\right] \triangleq \mathbb{E}\left[\nabla\ell(\widetilde{w} - \Delta, Z)\right]$ will enable us to have the bound below.

**Theorem 1.** *The generalization error of SGD is upper bounded by*

$$\sqrt{\frac{R^2 d}{n}\sum_{t=1}^{T}\mathbb{E}\left[\log\left(\frac{\lambda_t^2\mathbb{E}\left[||g(W_{t-1}, B_t) - \mathbb{E}\left[g(W_{t-1}, Z)\right]||^2\right]}{d\sigma_t^2} + 1\right)\right]} + |\mathbb{E}\left[\gamma(W_T, S) - \gamma(W_T, S')\right]|.$$

This bound is strictly tighter than the bound in Lemma 2. In fact, from Theorem 1, one can recover Lemma 2 with a smaller constant factor (see Appendix C.3 for more details).

To completely remove $\Psi(w_{t-1})$ and to obtain a closed form of the optimal bound, we will let $\Omega = \mathbb{E}\left[g(W, Z)\right]$, then Lemma 4 gives us a crisp way to have the following upper bound that is independent of the distribution of $\Delta$.

**Lemma 5.** *Let $G_t = -\lambda_t g(W_{t-1}, B_t)$. Then, $I(G_t + N_t; S|\widetilde{W}_{t-1}) \leq \frac{1}{2}d\log\left(\frac{\lambda_t^2\mathbb{E}[\mathbb{V}_t(W_{t-1})]}{d\sigma_t^2} + 1\right)$.*

In this lemma, the mutual information $I(G_t + N_t; S|\widetilde{W}_{t-1})$ roughly indicates the degree by which the SGD's updating signal $G_t$ (smoothed with noise) depends on the training sample $S$, when $B_t$ is used for computing the gradient. When this dependency is strong (giving rise to a high value of the mutual information), the model conceivably tends to overfit the training sample. This lemma suggests that the strength of this dependency can be upper-bounded by the expected gradient dispersion at the current weight configuration. In our experiments, we will estimate the expected gradient dispersion and validate this intuition.

We are now in a position to state our main theorem.

**Theorem 2.** *The generalization error of SGD is upper bounded by*

$$|\mathrm{gen}(\mu, P_{W_T|S})| \leq \sqrt{\frac{R^2 d}{n}\sum_{t=1}^{T}\log\left(\frac{\lambda_t^2\mathbb{E}\left[\mathbb{V}_t(W_{t-1})\right]}{d\sigma_t^2} + 1\right)} + |\mathbb{E}\left[\gamma(W_T, S) - \gamma(W_T, S')\right]|. \quad (2)$$

*Further, assume $L_\mu(w_T) \leq \mathbb{E}_\Delta [L_\mu(w_T + \Delta_T)]$, $\ell$ is twice differentiable, and $\sigma_t^2$ is independent of t. Denote by $\mathrm{H}_{W_T}$ the Hessian of the loss with respect to $W_T$ and let $\mathrm{Tr}(\cdot)$ denote trace. Then*

$$\mathrm{gen}(\mu, P_{W_T|S}) \leq \frac{3}{2} \left( \sum_{t=1}^{T} \frac{R^2 \lambda_t^2 T}{n} \mathbb{E}\left[\mathbb{V}_t(W_{t-1})\right] \mathbb{E}\left[\mathrm{Tr}\left(\mathrm{H}_{W_T}(Z)\right)\right] \right)^{\frac{1}{3}}. \tag{3}$$

**Remark 1.** *With a single draw of S and under the deterministic setting (i.e., with fixed weight initialization and batching trajectory), removing $\Psi(W_{t-1})$ will make the bound much tighter as shown in Figure 1. This remarkable improvement should come at no surprise, since $\Psi(W_{t-1})$ monotonically increases with training epochs (noting that $\Psi(W_{t-1})$ has the cumulative variance $\sum_{i=1}^{t-1} \sigma_i^2 \mathrm{I}_d$ ) while $\mathbb{V}_t(W_{t-1})$ appears decreasing (see Figure 3a). Figure 1 also indicates that if the noise variance $\sigma_t$ is made small, one expects the gradient dispersion to dominate the trajectory term in Lemma 2. However, the factor $1/\sigma_t^2$ will become too large, make the bound loose. More discussions about the stochastic setting with multiple draws of S are deferred to Appendix E.2.*

**Remark 2.** *The condition $L_\mu(w_T) \leq \mathbb{E}_{\Delta_T} [L_\mu(w_T + \Delta_T)]$ indicates that the perturbation does not decrease the population risk. This is also assumed in Foret et al. (2020) in the derivation of a PAC-Bayesian generalization bound.*

In the bound of Eq.2, the first term captures the impact of the training trajectory ("trajectory term"), and the second term captures the impact of the final solution, which in fact measures the flatness for the loss landscape at the found solution ("flatness term"). The learning rate in SGD has explicitly appeared in the trajectory term of Eq.3. From the way it appears in the bound, one may be tempted to assert that a small learning rate will improve generalization. This would then contradict some previous observations (Jastrzebski et al., 2017; Wu et al., 2018; He et al., 2019), in which large learning rate will benefit generalization. In addition, the bound also suggests that the batch size has some direct impact on the trajectory term through gradient dispersion. We investigate this by performing experiments with varying learning rates and batch sizes (see Figure 6 in Appendix E.3). As seen in the experiments, increasing the learning rate impacts the trajectory term and the flatness term in opposite ways, i.e. making one increase and the other decrease. A similar (but reverted) behaviour is also observed with batch sizes. This makes the generalization bound in Eq.3, have a rather complex relationship with the settings of learning rate and batch size.

Eq.3 follows from Eq.2 by minimizing the bound over $\sigma$. The bound in Eq.3 is thus independent of a choice of $\sigma_t$'s and can be computed easily and efficiently.

To summarize, we remark that these bounds suggest that in order for the model to generalize well, both the trajectory term and the flatness term need to be small — the former involves the learning rate with the gradient dispersion along the training trajectory, whereas the latter depends on the flatness of the empirical risk surface at the found solution.

## 4 APPLICATION IN LINEAR NETWORKS AND TWO-LAYER RELU NETWORKS.

We now apply Theorem 2 to two neural network models in a regression setting. Let $Z = (X, Y)$ with $X \in \mathbb{R}^{d_0}$ and $Y \in \mathbb{R}$. Assume $||X|| = 1$. We use SGD to train a model $f(W, \cdot) : \mathbb{R}^{d_0} \to \mathbb{R}$ and define the loss as $\ell(W, Z) = 1/2(Y - f(W, X))^2$. The bounds below are both in the form of the product of the flatness term and the trajectory term.

**Theorem 3** (Linear Networks). *Let the model be a linear network, i.e. $f(W, X) = W^T X$, then*

$\mathrm{gen}(\mu, P_{W_T|S}) \leq 3 \left( \sum_{t=1}^{T} \frac{R^2 \lambda_t^2 T}{4n} \mathbb{E}\left[\ell(W_{t-1}, Z)\right] \right)^{\frac{1}{3}}$.

**Theorem 4** (Two-Layer ReLU Networks). *Following Arora et al. (2019), consider $f(W, X) = \frac{1}{\sqrt{m}} \sum_{r=1}^{m} A_r \mathrm{ReLU}(W_r^T X)$ where m is the width of the neural network, $A_r \sim \mathrm{unif}(\{+1, -1\})$ and $\mathrm{ReLU}(\cdot)$ is the ReLU activation. We only train the first layer parameters $W = [W_1, W_2, \ldots, W_m] \in \mathbb{R}^{d_0 \times m}$ and fix the second layer parameters $A = [A_1, A_2, \ldots, A_m] \in \mathbb{R}^m$ during training. Then,*

$$\mathrm{gen}(\mu, P_{W_T|S}) \leq 3 \left( \sum_{r=1}^{m} \mathbb{E}\left[\frac{\mathbb{I}_{r,i,T}}{m}\right] \sum_{t=1}^{T} \frac{R^2 \lambda_t^2 T}{4n} \mathbb{E}\left[\sum_{r=1}^{m} \frac{\mathbb{I}_{r,i,t}}{m} \ell(W_{t-1}, Z)\right] \right)^{\frac{1}{3}},$$

*where $\mathbb{I}_{r,i,t} = \mathbb{I}\{W_{t-1,r}^T X_i \geq 0\}$ and $\mathbb{I}$ is the indicator function.*

Compared with Theorem 3, we notice that in the two-layer ReLU network, the ReLU activation state along the training trajectory plays a key role in the bound of Theorem 4. Specifically, the weights of the deactivated neurons do not contribute to the bound of Theorem 4, making the bound not explicitly depend on the model dimension $d$. This result also suggests that sparsely activated ReLU networks are expected to generalize better. Despite various empirical evidence pointing to this behaviour (see, e.g., Glorot et al. (2011)), to the best of our knowledge, this theorem provides the first theoretical justification in this regard.

The comparison between these bounds and the dynamics of generalization gap is deferred to Appendix E.4. In the remainder of this paper, we will study more complex network architectures beyond the linear and two-layer ReLU networks and implications of our bounds in those settings.

## 5 EXPERIMENTAL STUDY

**Bound Verification**  We first verify our bound of Eq.3 in Theorem 2 by training an MLP (with one hidden layer) and an AlexNet (Krizhevsky et al., 2012) on MNIST and CIFAR10 (Krizhevsky, 2009), respectively. To simplify estimation, we fix the weight initialization and set $\sigma_t$ and $\lambda_t$ to be constants $\sigma$ and $\lambda$, respectively. To compute $\sum_{t=1}^{T} \mathbb{E}\left[\mathbb{V}_t(W_{t-1})\right]$, we compute the gradient dispersion as its empirical estimate from a batch, utilizing a PyTorch (Paszke et al., 2019) library BackPack (Dangel et al., 2020). To compute $\text{Tr}\left(\mathbb{E}\left[\text{H}_{W_T}(Z)\right]\right)$, we use the PyHessian library (Yao et al., 2020) to compute the Hessian.

We perform experiments with varying network width and varying levels of label noise. Specifically, noise level $\epsilon$ refers to the setting where we replace the labels of $\epsilon$ fraction of the training and testing instances with random labels. The estimated bound is compared against the true generalization gap, namely, the difference between the training loss and testing loss, and is shown in Figure 2.

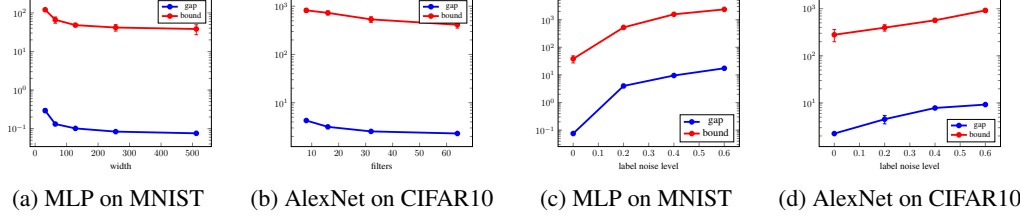

| (a) MLP on MNIST | (b) AlexNet on CIFAR10 | (c) MLP on MNIST | (d) AlexNet on CIFAR10 |

Figure 2: Estimated bound and empirical generalization gap ("gap") as functions of network width ((a) and (b)) and label noise level ((c) and (d)). Note that Y-axis is in log scale.

In Figure 2, we see that in all cases the estimated bound follows closely the trend of the true generalization gap. The fact that the bound curve consistently tracks the gap curve under various label noise levels indicates that our bound very well captures the changes of the data distribution. Note that in Figure 2 (a) and (b), our bound decays with the increase of the model size, showing a trend as opposite to the bounds obtained in classical learning theory. But such a trend clearly better explains the generalization behaviour of modern neural networks.

**Epoch-wise Double Descent of Gradient Dispersion**  We experimentally investigate the impact of gradient dispersion on the training of the neural networks by fixing the learning rate, batch size and weight initialization for the each model (MLP for MNIST, AlexNet for CIFAR10). For each model and various label noise levels, we plot in Figure 3 the evolution of the (empirical) gradient dispersion $\widehat{\mathbb{V}}_t(w_{t-1})$, training accuracy and testing accuracy across training epochs.

An intriguing epoch-wise "double descent" phenomenon is observed, particularly when the labels are noisy. According to the double descent curve, the training may be split into three phases (e.g., Figure 3 (h)). In the first phase, the gradient dispersion rapidly descends and maintains a very low level. In this phase, both training and test accuracies increase while maintaining a very small generalization gap. This suggests that the network in this phase is extracting useful patterns and generalizes well. In the second phase, the gradient dispersion starts increasing until it reaches a

peak value. In this phase, the training and testing accuracies gradually diverge, marking the model entering an overfitting or "memorization" regime – when the data contains the noisy labels, the network mostly tries to memorize the labels in the training set. In the third phase, the gradient dispersion descends again, reaching a low value. In this phase, the model continuously overfits the training data, until the training and testing curves reach their respective maximum and minimum. It appears that the timing of the three phases depends on the dataset and the label noise level. For simpler data (e.g. MNIST) and cleaner datasets, the first phase may be shorter. This is arguably because in these datasets, extracting useful patterns is relatively easier. Nonetheless, the valley in the double-descent curve appears to mark a "great divide" between generalization and memorization.

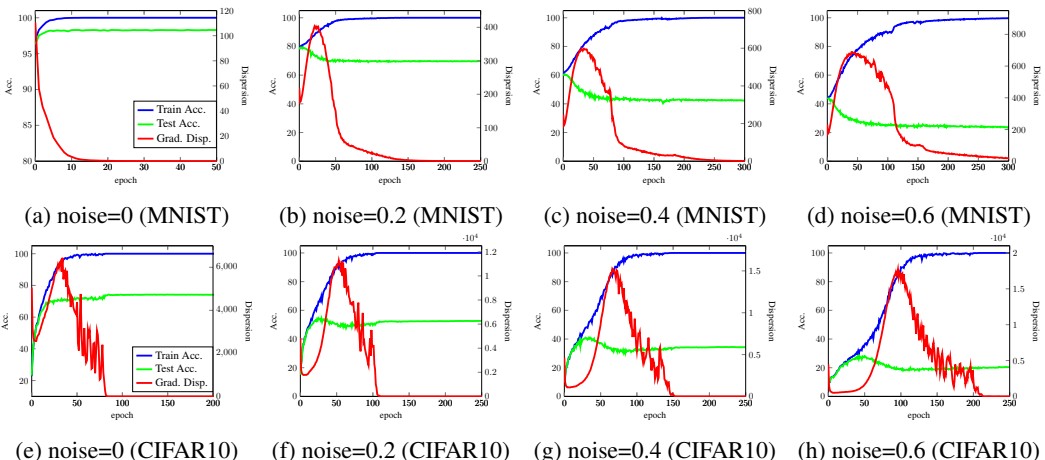

(a) noise=0 (MNIST)  (b) noise=0.2 (MNIST)  (c) noise=0.4 (MNIST)  (d) noise=0.6 (MNIST)

(e) noise=0 (CIFAR10)  (f) noise=0.2 (CIFAR10)  (g) noise=0.4 (CIFAR10)  (h) noise=0.6 (CIFAR10)

Figure 3: Dynamics of gradient dispersion, in relation to training/testing accuracies.

**Dynamic Gradient Clipping**  Inspired by our generalization bounds and above observations, one way to reduce the generalization error is to control the trajectory term of the bounds by reducing the gradient dispersion in each training step. Here we investigate a simple scheme that dynamically clips the gradient norm so as to reduce the gradient dispersion. Specifically, whenever the current gradient norm is larger than the gradient norm $K$ steps earlier, or $||g(W_t, B_t)||_2 > ||g(W_{t-K}, B_{t-K})||_2$ (i.e., the model is expected to have entered the "memorization" regime), we reduce the norm of the current gradient $g(W_t, B_t)$ to $\alpha$ fraction of $||g(W_{t-K}, B_{t-K})||_2$, for some prescribed value $\alpha < 1$.

The effectiveness of this scheme is best demonstrated when the labels contain noise. As shown in Figure 4 (and Figure 9 in Appendix E.5), dynamic gradient clipping significantly closes the gap between the training accuracy and the testing accuracy. The models trained with this scheme maintain a near-optimal testing accuracy (e.g., about 80% when the label noise level of MNIST is 0.2), without suffering from the severe memorization effect as seen in models trained without this scheme. Further understanding of the double-descent phenomenon of the gradient dispersion

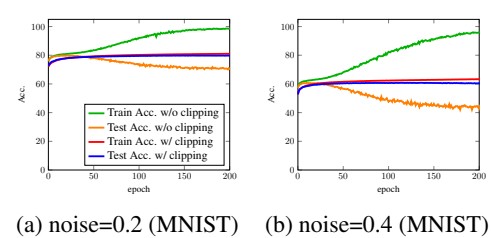

(a) noise=0.2 (MNIST)  (b) noise=0.4 (MNIST)

Figure 4: Dynamic Gradient Clipping.

may enable more delicate design of such a dynamic clipping scheme and potentially lead to novel and powerful regularization techniques.

## 6  A PRACTICAL IMPLICATION: GAUSSIAN MODEL PERTURBATION

The appearance of the flatness term in our generalization bounds suggests that for an empirical risk minimizer $w^*$ to generalize well, it is necessary that the empirical risk surface at $w^*$ is flat, or insensitive to a small perturbation of $w^*$. This naturally motivates a training scheme using the

following regularized loss:

$$\min_w L_s(w) + \rho \mathop{\mathbb{E}}_{\Delta \sim \mathcal{N}(0, \sigma^2 \mathbf{I}_d)} [L_s(w + \Delta) - L_s(w)],$$

where $\rho$ is a hyper-parameter. Replacing the expectation above with its stochastic approximation using $k$ realizations of $\Delta$ gives rise to the following optimization problem.

$$\min_w \frac{1}{b} \sum_{z \in B} \left( (1 - \rho)\ell(w, z) + \rho \frac{1}{k} \sum_{i=1}^{k} (\ell(w + \delta_i, z)) \right).$$

We refer to the SGD training scheme using this loss as *Gaussian model perturbation* or GMP. Notably, GMP requires $k + 1$ forward passes for every parameter update. Empirical evidence shows that a small $k$, for example, $k = 3$, already gives competitive performance. Implementing the $k + 1$ forward passes on parallel processors further reduces the computation load.

| Method | SVHN | CIFAR-10 | CIFAR-100 |
|--------|------|----------|-----------|
| ERM | 96.86±0.060 | 93.68±0.193 | 72.16±0.297 |
| Dropout | 97.04±0.049 | 93.78±0.147 | 72.28±0.337 |
| L. S. | 96.93±0.070 | 93.71±0.158 | 72.51±0.179 |
| Flooding | 96.85±0.085 | 93.74±0.145 | 72.07±0.271 |
| MixUp | 96.91±0.057 | **94.52±0.112** | 73.19±0.254 |
| Adv. Tr. | 97.06±0.091 | 93.51±0.130 | 70.88±0.145 |
| AMP | **97.27±0.015** | 94.35±0.147 | 74.40±0.168 |
| **GMP**[3] | 97.18±0.057 | 94.33±0.094 | 74.45±0.256 |
| **GMP**[10] | 97.09±0.068 | 94.45±0.158 | **75.09±0.285** |

Table 1: Top-1 classification accuracy acc.(%) of VGG16. We run experiments 10 times and report the mean and the standard deviation of the testing accuracy. Superscript denotes the number of sampled Gaussian noises during training.

We compare GMP with several major regularization schemes in the current art, including Dropout (Srivastava et al., 2014), label smoothing (L.S.) (Szegedy et al., 2016), Flooding (Ishida et al., 2020), MixUp (Zhang et al., 2018), adversarial training (Adv. Tr.) (Goodfellow et al., 2015), and AMP (Zheng et al., 2021). We will also include ERM as the baseline, where no regularization method applied. The compared schemes are evaluated on three popular benchmark image classification datasets SVHN (Netzer et al., 2011), CIFAR-10 and CIFAR-100 (Krizhevsky, 2009). Two representative deep architectures VGG16 (Simonyan & Zisserman, 2015) and PreActResNet18 (He et al., 2016) are taken as the underlying model. We train the models for 200 epochs by SGD. The learning rate is initialized as 0.1 and divided by 10 after 100 and 150 epochs. For all compared models, the batch size is set to 50 and weight decay is set to $10^{-4}$. For GMP, we choose $\rho = 0.5$ and set the standard deviation of the Gaussian noise $\Delta$ to 0.03. The value of $k$ is chosen as 3 and 10 respectively (referred to as GMP[3] and GMP[10]).

The performances of all compared schemes are given in Table 1 (and the results of PreActResNet18 are shown in Appendix E.7). For the compared regularization schemes except GMP, we directly report their performances as given in Zheng et al. (2021). Table 1 demonstrates the effectiveness of GMP. Overall GMP performs comparably to the current art of regularization schemes, although appearing slightly inferior to the most recent record given by AMP on SVHN and MixUp on CIFAR-10, respectively (Zheng et al., 2021). Noting that the key ingredient of AMP, "max-pooling" in the parameter space, greatly resembles regularization term in GMP, which may be seen as "average-pooling" in the same space.

## 7 CONCLUSION AND OUTLOOK

This paper presents new information theoretic generalization bounds for models (e.g., linear networks and two-layer ReLU neural networks) trained with SGD. Our bounds naturally point to new and effective regularization schemes. At the same time, our bounds and experimental study reveal interesting phenomena in the SGD training of neural networks. There are yet promising directions for further improving these bounds, for example, via exploiting conditional mutual information bounds (Haghifam et al. (2020)), strong data processing inequalities (Wang et al. (2021a)), and the relationship between between the trajectory term and the flatness term.

ACKNOWLEDGMENTS

This work is supported partly by an NSERC Discovery Grant and Artificial Intelligence for Design Challenge program of National Research Council Canada. The authors would like to sincerely thank Gergely Neu and Borja Rodríguez-Gálvez for pointing out errors in the previous version of this paper. The authors would also like to thank Maia Fraser for stimulating discussions.

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

# Appendix

## Table of Contents

## A  HWI INEQUALITY: PROOF OF LEMMA 3

The proof given here is a simple extension of the proof of Raginsky & Sason (2018, Lemma 3.4.2), which is a special instance of the (weak) HWI inequality.

*Proof.*

$$
\begin{aligned}
\mathrm{D}_{\mathrm{KL}}(P_{X+\sqrt{t}N}||P_{Y+\sqrt{t'}N}) \leq & \mathrm{D}_{\mathrm{KL}}(P_{X,Y,X+\sqrt{t}N}||P_{X,Y,Y+\sqrt{t'}N}) & (1) \\
= & \underset{X,Y}{\mathbb{E}}\left[\mathrm{D}_{\mathrm{KL}}(P_{X+\sqrt{t}N|X,Y}||P_{Y+\sqrt{t'}N|X,Y})\right] \\
= & \underset{X,Y}{\mathbb{E}}\left[\mathrm{D}_{\mathrm{KL}}(\mathcal{N}(X,t\mathrm{I}_d)||\mathcal{N}(Y,t'\mathrm{I}_d))\right] & (2) \\
= & \frac{1}{2t'}\underset{X,Y}{\mathbb{E}}\left[||X-Y||^2\right] + \frac{d}{2}(\log\frac{t'}{t}+\frac{t}{t'}-1), & (3)
\end{aligned}
$$

where Eq.1 is by the chain rule of the KL divergence, Eq.2 holds since $N$ is independent of $(X,Y)$ and Eq.3 is by the equality

$$
\mathrm{D}_{\mathrm{KL}}(\mathcal{N}(\mu_1,\Sigma_1)||\mathcal{N}(\mu_2,\Sigma_2)) = \frac{1}{2}\left[\log\frac{|\Sigma_2|}{|\Sigma_1|} - d + \mathrm{Tr}\left(\Sigma_2^{-1}\Sigma_1\right) + (\mu_2-\mu_1)^T\Sigma_2^{-1}(\mu_2-\mu_1)\right].
$$

Eq.3 holds for any joint distribution of $(X, Y)$. If we let $t' = t$, by the definition of Wasserstein distance, we have

$$\mathrm{D}_{\mathrm{KL}}(P_{X+\sqrt{t}N}||P_{Y+\sqrt{t}N}) \leq \frac{1}{2t}\mathbb{W}_2^2(P_X, P_Y).$$

This recovers the vector version of Raginsky & Sason (2018, Lemma 3.4.2). $\qquad\square$

## B  A KEY LEMMA: LEMMA 4

*Proof.* We give two proofs of this lemma, the first proof is inspired by the techniques used in Wang et al. (2021a) and Guo et al. (2005), and the second proof is similar to Pensia et al. (2018).

1. Let $Z = Y + \Delta$. By the properties of mutual information,

$$I(f(Z, X) + \sigma N; X|Y = y) = I(f(Z, X) - \Omega + \sigma N; X|Y = y).$$

Let $g(Z, X) \triangleq f(Z, X) - \Omega$. Let $P_{N'} = \mathcal{N}(0, \sigma'^2 \mathbf{I}_d)$ for any $\sigma' > 0$ and $N'$ is independent of other random variables for a fixed $y$. Then, for $Y = y$,

$$I(g(Z, X) + \sigma N; X|Y = y) = \underset{X}{\mathbb{E}}\left[\mathrm{D}_{\mathrm{KL}}\left(P_{g(Z,x)+\sigma N|X=x,Y=y}||P_{g(Z,X)+\sigma N|Y=y}\right)\right]$$

$$= \underset{X}{\mathbb{E}}\left[\mathrm{D}_{\mathrm{KL}}\left(P_{g(Z,x)+\sigma N|X=x,Y=y}||P_{N'|Y=y}\right) - \mathrm{D}_{\mathrm{KL}}\left(P_{g(Z,X)+\sigma N|Y=y}||P_{N'|Y=y}\right)\right]$$
(4)

$$\leq \inf_{\sigma'>0} \underset{X}{\mathbb{E}}\left[\mathrm{D}_{\mathrm{KL}}\left(P_{g(Z,x)+\sigma N|X=x,Y=y}||P_{\sigma'N|Y=y}\right)\right] \quad (5)$$

$$= \inf_{\sigma'>0} \frac{1}{2\sigma'^2}\underset{X}{\mathbb{E}}\left[\underset{\Delta}{\mathbb{E}}\left[||g(y+\Delta, x)||^2|X=x, Y=y\right]\right] + \frac{d}{2}(\log\frac{\sigma'^2}{\sigma^2} + \frac{\sigma^2}{\sigma'^2} - 1)$$
(6)

$$= \frac{d}{2}\log\left(\frac{\underset{X,\Delta}{\mathbb{E}}\left[||g(y+\Delta, X)||^2|Y=y\right]}{d\sigma^2} + 1\right),$$

where Eq. 4 is the golden formula (see Theorem 3.3 in Polyanskiy & Wu (2019) ), Eq. 5 is by the non-negativity of relative entropy, Eq. 6 is due to Lemma 3 and the last equality holds when the optimal

$$\sigma'^2 = \frac{\underset{X,\Delta}{\mathbb{E}}\left[||g(y+\Delta, X)||^2|Y=y\right]}{d} + \sigma^2$$

is achieved.

Thus,

$$I(f(Z, X) + N; X|Y) = \underset{Y}{\mathbb{E}}\left[I(f(Z, X) + N; X|Y = y)\right]$$

$$\leq \underset{Y}{\mathbb{E}}\left[\frac{d}{2}\log\left(\frac{\underset{X,\Delta}{\mathbb{E}}\left[||g(y+\Delta, X)||^2|Y=y\right]}{d\sigma^2} + 1\right)\right]$$

$$= \underset{Y}{\mathbb{E}}\left[\frac{d}{2}\log\left(\frac{\underset{X,\Delta}{\mathbb{E}}\left[||f(Z, X) - \Omega||^2|Y=y\right]}{d\sigma^2} + 1\right)\right],$$

which completes the proof.

2. Alternatively, let $h(\cdot)$ denote the differential entropy, then

$$
\begin{aligned}
I(f(Z,X) + \sigma N; X|Y = y) &= I(g(Z,X) + \sigma N; X|Y = y) \\
&= h(g(Z,X) + \sigma N|Y = y) - h(g(Z,X) + \sigma N|X, Y = y) \\
&\leq h(g(Z,X) + \sigma N|Y = y) - h(g(Z,X) + \sigma N|\Delta, X, Y = y) \\
&\qquad\qquad\qquad\qquad\qquad\qquad\qquad\qquad\qquad\qquad (7) \\
&= h(g(Z,X) + \sigma N|Y = y) - h(\sigma N) \\
&= h(g(Z,X) + \sigma N|Y = y) - \frac{d}{2} \log 2\pi e \sigma^2,
\end{aligned}
$$

where Eq. 7 is due to the fact that conditioning reduces entropy.

Notice that

$$
\underset{\Delta, X, N}{\mathbb{E}} \left[ ||g(y + \Delta, X) + \sigma N||^2|Y = y \right] = \underset{\Delta, X}{\mathbb{E}} \left[ ||g(y + \Delta, X)||^2|Y = y \right] + d\sigma^2.
$$

Since the Gaussian distribution maximizes the entropy over all distributions with the same variance (Cover & Thomas, 2012), we have

$$
h(g(Z,X) + \sigma N|Y = y) \leq \frac{d}{2} \log 2\pi e \frac{\underset{\Delta, X}{\mathbb{E}} \left[ ||g(y + \Delta, X)||^2|Y = y \right] + d\sigma^2}{d}.
$$

Therefore,

$$
\begin{aligned}
I(f(Z,X) + \sigma N; X|Y = y) &\leq \frac{d}{2} \log 2\pi e \frac{\underset{\Delta, X}{\mathbb{E}} \left[ ||g(y + \Delta, X)||^2|Y = y \right] + d\sigma^2}{d} - \frac{d}{2} \log 2\pi e \sigma^2 \\
&= \frac{d}{2} \log \left( \frac{\underset{\Delta, X}{\mathbb{E}} \left[ ||g(y + \Delta, X)||^2|Y = y \right]}{d\sigma^2} + 1 \right).
\end{aligned}
$$

The remaining part is straightforward and is the same with the previous proof.

$\square$

# C  Mutual Information Bounds for SGD: Proof of Theorem 1 and Theorem 2

In this section, we provide a proof for Theorem 1 and Theorem 2, while elaborating on why the local gradient sensitivity term in Lemma 2 can be removed. The key ingredient that makes this possible is a construction given in the proof of Lemma 5.

## C.1  Lemma 6

We first give the following useful lemma.

**Lemma 6.** *The mutual information* $I(\widetilde{W}_T; S) \leq \sum_{t=1}^{T} I\left(-\lambda_t g(W_{t-1}, B_t) + N_t; S|\widetilde{W}_{t-1}\right).$

*Proof.* The mutual information between the final output of SGD and the training sample can be upper bounded by the mutual information between the full trajectories and the training sample,

which is shown below.

$$I(\widetilde{W}_T; S)$$

$$=I\left(\widetilde{W}_{T-1} - \lambda_T g(W_{T-1}, B_T) + N_T; S\right)$$

$$\leq I\left(\widetilde{W}_{T-1}, -\lambda_T g(W_{T-1}, B_T) + N_T; S\right) \tag{8}$$

$$=I(\widetilde{W}_{T-1}; S) + I\left(-\lambda_T g(W_{T-1}, B_T) + N_T; S|\widetilde{W}_{T-1}\right) \tag{9}$$

$$\leq I(\widetilde{W}_{T-2}; S) + I\left(-\lambda_{T-1} g(W_{T-2}, B_{T-1}) + N_{T-1}; S|\widetilde{W}_{T-2}\right) + I\left(-\lambda_T g(W_{T-1}, B_T) + N_T; S|\widetilde{W}_{T-1}\right) \tag{10}$$

$$\vdots$$

$$\leq I(\widetilde{W}_0; S) + \sum_{t=1}^{T} I\left(-\lambda_t g(W_{t-1}, B_t) + N_t; S|\widetilde{W}_{t-1}\right), \tag{11}$$

$$= \sum_{t=1}^{T} I\left(-\lambda_t g(W_{t-1}, B_t) + N_t; S|\widetilde{W}_{t-1}\right), \tag{12}$$

where Eq.8 is by $I(f(X,Y); Z) \leq I(X,Y; Z)$ (i.e. $Z - (X,Y) - f(X,Y)$ forms a Markov chain and then use the data processing inequality), Eq.9 is by the chain rule of mutual information, Eq.10 is by applying the similar procedure (namely, Eq.8-Eq.9) to $I(\widetilde{W}_{T-1}; S)$, Eq. 11 is by doing these steps recursively and Eq.12 is due to the fact that $\widetilde{W}_0$ is independent of $S$ (i.e. $I(\widetilde{W}_0; S) = 0$). $\square$

## C.2 PROOF OF THEOREM 1

*Proof.* We first follow the similar decomposition of the expected generalization error and apply Lemma 1 as in Neu et al. (2021),

$$|\text{gen}(\mu, P_{W_T|S})| = \left| \text{gen}(\mu, P_{\widetilde{W}_T|S}) + \underset{W_T, \Delta_T}{\mathbb{E}}\left[L_\mu(W_T) - L_\mu(\widetilde{W}_T)\right] + \underset{W_T, \Delta_T, S}{\mathbb{E}}\left[L_S(\widetilde{W}_T) - L_S(W_T)\right]\right|$$

$$\leq \sqrt{\frac{2R^2}{n} I(\widetilde{W}_T; S)} + \left|\underset{W_T, S, S'}{\mathbb{E}}\left[\gamma(W_T, S) - \gamma(W_T, S')\right]\right|. \tag{13}$$

The remaining task is to bound the mutual information $I(\widetilde{W}_T; S)$.

Let $X = S, Y = \widetilde{W}_{t-1}$ and $\Omega = \underset{\Delta_{t-1}, Z}{\mathbb{E}}[g(\widetilde{w}_{t-1} - \Delta_{t-1}, Z)]$, by applying Lemma 4, we have

$$I\left(-\lambda_t g(W_{t-1}, B_t) + N_t; S|\widetilde{W}_{t-1}\right)$$

$$\leq \underset{\widetilde{W}_{t-1}}{\mathbb{E}}\left[\frac{d}{2}\log\left(\frac{\lambda_t^2 \underset{S, \Delta_{t-1}}{\mathbb{E}}\left[||g(\widetilde{w}_{t-1} - \Delta_{t-1}, B_t) - \underset{\Delta_{t-1}, Z}{\mathbb{E}}[g(\widetilde{w}_{t-1} - \Delta_{t-1}, Z)]||^2|\widetilde{W}_{t-1} = \widetilde{w}_{t-1}\right]}{d\sigma^2} + 1\right)\right].$$

By Lemma 6 and putting everything together, we have

$$|\text{gen}(\mu, P_{W_T|S})|$$

$$\leq \sqrt{\frac{R^2 d}{n} \sum_{t=1}^{T} \underset{\widetilde{W}_{t-1}}{\mathbb{E}}\left[\log\left(\frac{\lambda_t^2 \underset{S, \Delta_{t-1}}{\mathbb{E}}\left[||g(W_{t-1}, B_t) - \underset{Z, \Delta_{t-1}}{\mathbb{E}}[g(W_{t-1}, Z)]||^2\right]}{d\sigma_t^2} + 1\right)\right]} + |\mathbb{E}\left[\gamma(W_T, S) - \gamma(W_T, S')\right]|.$$

This completes the proof.

$$\square$$

### C.3 COROLLARY 1: RECOVERING LEMMA 2

We can recover the result of Lemma 2 in Neu et al. (2021) from Theorem 1, which is re-stated in the following corollary.

**Corollary 1.** *The generalization error of SGD is upper bounded by*

$$|\mathrm{gen}(\mu, P_{W_T|S})| \le \sqrt{\frac{2R^2}{n} \sum_{t=1}^{T} \frac{\lambda_t^2}{\sigma_t^2} \mathop{\mathbb{E}}_{W_{t-1}} \left[ 3\Psi(W_{t-1}) + 2\widetilde{\mathbb{V}}_t(W_{t-1}) \right]} + \left| \mathbb{E}\left[ \gamma(W_T, S) - \gamma(W_T, S') \right] \right|.$$

*Proof.* From the first term in Theorem 1, we have

$$\sqrt{\frac{R^2 d}{n} \sum_{t=1}^{T} \mathop{\mathbb{E}}_{\widetilde{W}_{t-1}} \left[ \log\left( \frac{\lambda_t^2 \mathop{\mathbb{E}}_{S,\Delta_{t-1}} \left[ \left\| g(W_{t-1}, B_t) - \mathop{\mathbb{E}}_{Z,\Delta_{t-1}} [g(W_{t-1}, Z)] \right\|^2 \right]}{d\sigma_t^2} + 1 \right) \right]}$$

$$\le \sqrt{\frac{R^2 d}{n} \sum_{t=1}^{T} \log\left( \frac{\lambda_t^2 \mathop{\mathbb{E}}_{\widetilde{W}_{t-1},\Delta_{t-1},S} \left[ \left\| g(W_{t-1}, B_t) - \mathop{\mathbb{E}}_{Z,\Delta_{t-1}} \left[ g(\widetilde{W}_{t-1} - \Delta_{t-1}, Z) \right] \right\|^2 \right]}{d\sigma_t^2} + 1 \right)} \tag{14}$$

$$\le \sqrt{\frac{R^2}{n} \sum_{t=1}^{T} \frac{\lambda_t^2}{\sigma_t^2} \mathop{\mathbb{E}}_{\widetilde{W}_{t-1},\Delta_{t-1},S} \left[ \left\| g(W_{t-1}, B_t) - \mathop{\mathbb{E}}_{Z,\Delta_{t-1}} \left[ g(\widetilde{W}_{t-1} - \Delta_{t-1}, Z) \right] \right\|^2 \right]} \tag{15}$$

$$= \sqrt{\frac{R^2}{n} \sum_{t=1}^{T} \frac{\lambda_t^2}{\sigma_t^2} \mathop{\mathbb{E}}_{W_{t-1},S} \left[ \left\| g(W_{t-1}, B_t) - \mathop{\mathbb{E}}_{Z} \left[ g(\widetilde{W}_{t-1}, Z) \right] + \mathop{\mathbb{E}}_{Z} \left[ g(\widetilde{W}_{t-1}, Z) \right] - \mathop{\mathbb{E}}_{Z,\Delta_{t-1}} \left[ g(\widetilde{W}_{t-1} - \Delta_{t-1}, Z) \right] \right\|^2 \right]}$$

$$\le \sqrt{\frac{R^2}{n} \sum_{t=1}^{T} \frac{\lambda_t^2}{\sigma_t^2} \mathop{\mathbb{E}}_{W_{t-1},S} \left[ 2 \left\| g(W_{t-1}, B_t) - \mathop{\mathbb{E}}_{Z} \left[ g(\widetilde{W}_{t-1}, Z) \right] \right\|^2 + 2 \left\| \mathop{\mathbb{E}}_{Z} \left[ g(\widetilde{W}_{t-1}, Z) \right] - \mathop{\mathbb{E}}_{Z,\Delta_{t-1}} \left[ g(\widetilde{W}_{t-1} - \Delta_{t-1}, Z) \right] \right\|^2 \right]} \tag{16}$$

$$\le \sqrt{\frac{R^2}{n} \sum_{t=1}^{T} \frac{\lambda_t^2}{\sigma_t^2} \mathop{\mathbb{E}}_{W_{t-1},S} \left[ 2 \left\| g(W_{t-1}, B_t) - \mathop{\mathbb{E}}_{Z} \left[ g(\widetilde{W}_{t-1}, Z) \right] \right\|^2 + 2 \mathop{\mathbb{E}}_{\Delta_{t-1}} \left[ \left\| \mathop{\mathbb{E}}_{Z} \left[ g(\widetilde{W}_{t-1}, Z) \right] - \mathop{\mathbb{E}}_{Z} \left[ g(\widetilde{W}_{t-1} - \Delta_{t-1}, Z) \right] \right\|^2 \right] \right]} \tag{17}$$

$$= \sqrt{\frac{R^2}{n} \sum_{t=1}^{T} \frac{\lambda_t^2}{\sigma_t^2} \mathop{\mathbb{E}}_{W_{t-1},S} \left[ 2 \left\| g(W_{t-1}, B_t) - \mathop{\mathbb{E}}_{Z} [g(W_{t-1}, Z)] + \mathop{\mathbb{E}}_{Z} [g(W_{t-1}, Z)] - \mathop{\mathbb{E}}_{Z} \left[ g(\widetilde{W}_{t-1}, Z) \right] \right\|^2 + 2\Psi(W_{t-1}) \right]}$$

$$\le \sqrt{\frac{R^2}{n} \sum_{t=1}^{T} \frac{\lambda_t^2}{\sigma_t^2} \mathop{\mathbb{E}}_{W_{t-1},S} \left[ 4 \left\| g(W_{t-1}, B_t) - \mathop{\mathbb{E}}_{Z} [g(W_{t-1}, Z)] \right\|^2 + 4 \left\| \mathop{\mathbb{E}}_{Z} [g(W_{t-1}, Z)] - \mathop{\mathbb{E}}_{Z} \left[ g(\widetilde{W}_{t-1}, Z) \right] \right\|^2 + 2\Psi(W_{t-1}) \right]} \tag{18}$$

$$= \sqrt{\frac{R^2}{n} \sum_{t=1}^{T} \frac{\lambda_t^2}{\sigma_t^2} \mathop{\mathbb{E}}_{W_{t-1},S} \left[ 4 \left\| g(W_{t-1}, B_t) - \mathop{\mathbb{E}}_{Z} [g(W_{t-1}, Z)] \right\|^2 + 6\Psi(W_{t-1}) \right]}$$

$$= \sqrt{\frac{2R^2}{n} \sum_{t=1}^{T} \frac{\lambda_t^2}{\sigma_t^2} \mathop{\mathbb{E}}_{W_{t-1}} \left[ 3\Psi(W_{t-1}) + 2\widetilde{\mathbb{V}}_t(W_{t-1}) \right]},$$

where Eq. 14 and 17 are by Jensen's inequality (i.e. the concavity of logarithm and the convexity of squared norm), Eq. 15 is by $\log(x+1) \le x$, and Eq. 16 and 18 are by $||x+y||^2 \le 2||x||^2 + 2||y||^2$.

This completes the proof. □

### C.4 PROOF OF LEMMA 5

*Proof.* We first notice that

$$I(G_t + N_t; S|\widetilde{W}_{t-1}) = I(-\lambda_t g(W_{t-1}, B_t) + \sigma_t N; S|\widetilde{W}_{t-1}),$$

where $N \sim \mathcal{N}(0, \mathrm{I}_d)$.

Then let $X = S, Y = \widetilde{W}_{t-1}$ and $\Omega = \mathop{\mathbb{E}}\limits_{W_{t-1}, Z}[\nabla \ell(W_{t-1}, Z)]$, by applying Lemma 4, we have

$$
I(-\lambda_t g(W_{t-1}, B_t) + \sigma_t N; S|\widetilde{W}_{t-1}) \le \frac{d}{2} \mathop{\mathbb{E}}\limits_{\widetilde{W}_{t-1}} \left[ \log \left( \frac{\lambda_t^2 \mathop{\mathbb{E}}\limits_{S, \Delta_{t-1}} \left[ ||g(W_{t-1}, B_t) - \mathop{\mathbb{E}}\limits_{W_{t-1}, Z}[\nabla_w \ell(W_{t-1}, Z)]||^2 \right]}{d\sigma_t^2} + 1 \right) \right]
$$

$$
\le \frac{d}{2} \log \left( \frac{\lambda_t^2 \mathop{\mathbb{E}}\limits_{W_{t-1}}[\mathbb{V}_t(W_{t-1})]}{d\sigma_t^2} + 1 \right),
$$

where the second inequality is by Jensen's inequality.

This completes the proof. □

### C.5 PROOF OF THEOREM 2

*Proof.* Applying Lemma 6 and Lemma 5 and putting everything together, we have

$$
|\mathrm{gen}(\mu, P_{W_T|S})| \le \sqrt{\frac{R^2 d}{n} \sum_{t=1}^{T} \log \left( \frac{\lambda_t^2 \mathop{\mathbb{E}}\limits_{W_{t-1}}[\mathbb{V}_t(W_{t-1})]}{d\sigma_t^2} + 1 \right)} + |\mathbb{E}\left[\gamma(W_T, S) - \gamma(W_T, S')\right]|.
$$

Next, to handle the mismatch between the outputs of perturbed SGD and SGD, we apply Taylor expansion around $\Delta_T = \vec{0}$,

$$
\mathop{\mathbb{E}}\limits_{W_T, S, \Delta_T}[L_S(W_T + \Delta_T) - L_S(W_T)] = \frac{1}{n} \sum_{i=1}^{n} \mathop{\mathbb{E}}\limits_{W_T, Z_i, \Delta_T}[\ell(W_T + \Delta_T, Z_i) - \ell(W_T, Z_i)]
$$

$$
\approx \mathop{\mathbb{E}}\limits_{W_T, Z, \Delta_T} \left[ \langle \nabla_w \ell(W_T, Z), \Delta_T \rangle + \frac{1}{2} \Delta_T^T \mathrm{H}_{W_T}(Z) \Delta_T \right]
$$

$$
= \mathop{\mathbb{E}}\limits_{W_T, Z, \Delta_T} \left[ \frac{1}{2} \Delta_T^T \mathrm{H}_{W_T}(Z) \Delta_T \right] \tag{19}
$$

$$
= \frac{1}{2} \langle \mathop{\mathbb{E}}\limits_{W_T, Z}[\mathrm{H}_{W_T}(Z)], \mathop{\mathbb{E}}\limits_{\Delta_T}\left[\Delta_T \Delta_T^T\right] \rangle
$$

$$
= \frac{1}{2} \langle \mathop{\mathbb{E}}\limits_{W_T, Z}[\mathrm{H}_{W_T}(Z)], \mathrm{diag}(\sum_{t=1}^{T} \sigma_t^2) \rangle \tag{20}
$$

$$
= \frac{\sum_{t=1}^{T} \sigma_t^2}{2} \mathrm{Tr}(\mathop{\mathbb{E}}\limits_{W_T, Z}[\mathrm{H}_{W_T}(Z)]),
$$

where Eq.19 is by the zero mean of the perturbation, Eq.20 is by the independence of the coordinates of $\Delta_T$, $\langle \cdot, \cdot \rangle$ denotes the inner product of two matrices, $\mathrm{diag}(A)$ is the diagonal matrix with element $A$ and $\mathrm{Tr}(\cdot)$ is the trace of a matrix.

Under the condition $\mathbb{E}_{W_T, S'} \left[ \gamma(W_T, S') \right] \geq 0$, we now bound $\text{gen}(\mu, \text{P}_{\widetilde{W}_T|S})$ instead of its absolute value, $|\text{gen}(\mu, \text{P}_{\widetilde{W}_T|S})|$. With the inequality $\log(x+1) \leq x$, the following is straightforward,

$$
\text{gen}(\mu, \text{P}_{\widetilde{W}_T|S}) \leq \sqrt{\frac{R^2}{n} \sum_{t=1}^{T} \frac{\lambda_t^2}{\sigma_t^2} \mathbb{E}_{W_{t-1}} \left[ \mathbb{V}(W_{t-1}) \right]} + \mathbb{E}_{W_T, S, S'} \left[ \gamma(W_T, S) - \gamma(W_T, S') \right]
$$

$$
\leq \sqrt{\frac{R^2}{n} \sum_{t=1}^{T} \frac{\lambda_t^2}{\sigma_t^2} \mathbb{E}_{W_{t-1}} \left[ \mathbb{V}(W_{t-1}) \right]} + \mathbb{E}_{W_T, S} \left[ \gamma(W_T, S) \right]
$$

$$
= \sqrt{\frac{R^2}{n} \sum_{t=1}^{T} \frac{\lambda_t^2}{\sigma_t^2} \mathbb{E}_{W_{t-1}} \left[ \mathbb{V}(W_{t-1}) \right]} + \frac{\sum_{t=1}^{T} \sigma_t^2}{2} \text{Tr}(\mathbb{E}_{W_T, Z} \left[ \text{H}_{W_T}(Z) \right]).
$$

Since every choice of $\sigma$ gives a valid generalization bound. The optimal bound can be found by simply utilizing the fact $A/\sigma + \sigma^2 B \geq 3(A/2)^{2/3} B^{1/3}$ for any positive $A$ and $B$, where the equality is achieved by the optimal $\sigma$. Finally, rearranging the terms will complete the proof. $\square$

## C.6 COROLLARY 2

**Corollary 2.** *If the loss function is differentiable and $\beta$-smooth with respect to $w$, then,*

$$
|\text{gen}(\mu, P_{W_T|S})| \leq \sqrt{\frac{R^2 d}{n} \sum_{t=1}^{T} \log \left( \frac{\lambda_t^2 \mathbb{E}_{W_{t-1}} \left[ \mathbb{V}_t(W_{t-1}) \right]}{d\sigma_t^2} + 1 \right)} + \beta d \sum_{t=1}^{T} \sigma_t^2.
$$

*Proof.* Recall the smoothness implies $f(\mathbf{v}) \leq f(\mathbf{w}) + \langle \nabla f(\mathbf{w}), \mathbf{v} - \mathbf{w} \rangle + \frac{\beta}{2} ||\mathbf{v} - \mathbf{w}||^2$ for all $\mathbf{v}$ and $\mathbf{w}$. By the triangle inequality, we have

$$
|\mathbb{E} \left[ L_\mu(W_T) - L_\mu(W_T + \Delta_T) \right]| \leq |\mathbb{E} \left[ \langle \nabla_w \ell(W_T, Z), \Delta_T \rangle \right]| + \frac{\beta}{2} \mathbb{E} \left[ ||\Delta_T||^2 \right] = \frac{\beta d \sum_{t=1}^{T} \sigma_t^2}{2}
$$

Thus, we can see that $|\mathbb{E} \left[ L_\mu(W_T) - L_\mu(W_T + \Delta_T) \right]| + |\mathbb{E} \left[ L_S(W_T + \Delta_T) - L_S(W_T) \right]| \leq \beta d \sum_{t=1}^{T} \sigma_t^2$.

This completes the proof. $\square$

**Remark 3.** *In Corollary 2, we note that the dependence of $d$ in the bound results from the spherical Gaussian noise used in the construction of the weight process $\widetilde{W}_T$. It is possible to replace the spherical Gaussian with a Gaussian noise having a non-diagonal covariance that reflects the geometry of the loss landscape. With this replacement, the dimension $d$ in the flatness term will be replaced by the trace of $\sum_{t=1}^{T} \kappa_t$, where $\kappa_t$ is the covariance matrix of the noise added at step $t$. Please refer to Neu et al. (2021) for a similar development.*

# D  APPLICATION IN NEURAL NETWORKS: PROOF OF THEOREMS 3 AND 4

## D.1  LEMMA 7

**Lemma 7.** *Under the same conditions of Theorem 2, the generalization error of SGD is upper bounded by*

$$
\text{gen}(\mu, P_{W_T|S}) \leq \frac{3}{2} \left( \sum_{t=1}^{T} \frac{R^2 \lambda_t^2 T}{n} \mathbb{E} \left[ ||g(W_{t-1}, B_t)||_2^2 \right] \mathbb{E} \left[ \text{Tr} \left( \text{H}_{W_T}(Z) \right) \right] \right)^{\frac{1}{3}}
$$

*Proof.* The proof of this lemma follows the same steps in the proof of Theorem 2 except that we require a different use of Lemma 4 here.

Specially, we let $\Omega = 0$ in Lemma 4. The remaining steps are the same in the proof of Theorem 2 and should be straightforward. $\qquad\square$

**Remark 4.** *The bound in Lemma 7 is weaker than Eq. 3 in Theorem 2 as the centralized expected vector norm should be smaller than the original expected vector norm.*

### D.2 PROOF OF THEOREM 3

*Proof.* By Cauchy–Schwarz inequality, we have

$$\mathbb{E}\left[||g(W_{t-1}, B_t)||_2^2\right] = \mathbb{E}\left[\left|\left|\frac{1}{b}\sum_{Z_i \in B_t} \nabla_w \ell(W_{t-1}, Z_i)\right|\right|_2^2\right] \leq \mathbb{E}\left[||\nabla_w \ell(W_{t-1}, Z)||_2^2\right]. \quad (21)$$

Notice that $\nabla_w \ell(W, Z) = (W^T X - Y)X$. Let $\hat{Y} = f(W, X)$. Then,

$$||\nabla_w \ell(W_{t-1}, Z)||_2^2 = ||(W_{t-1}^T X - Y)X||_2^2 \leq ||W_{t-1}^T X - Y||_2^2 = (\hat{Y} - Y)^2 = 2\ell(W_{t-1}, Z). \quad (22)$$

For the Hessian matrix, it's easy to see that $\text{Tr}(H_{W_T}(Z)) = 1$.

Plugging everything into Lemma 7, we have

$$\text{gen}(\mu, P_{W_T|S}) \leq 3\left(\sum_{t=1}^{T} \frac{R^2 \lambda_t^2 T}{4n} \mathbb{E}_{W_{t-1}, Z}[\ell(W_{t-1}, Z)]\right)^{\frac{1}{3}}.$$

This completes the proof. $\qquad\square$

### D.3 PROOF OF THEOREM 4

*Proof.* Since $\nabla_{w_r} \ell(W, Z_i) = \frac{1}{\sqrt{m}} A_r (f(W, X_i) - Y_i) X_i \mathbb{I}_{r,i}$, where $\mathbb{I}_{r,i} = \mathbb{I}\{W_r^T X_i \geq 0\}$, we have

$$||\nabla_w \ell(W_{t-1}, Z)||_2^2 \quad (23)$$
$$= \sum_{r=1}^{m} ||\frac{1}{\sqrt{m}} A_r (f(W_{t-1}, X_i) - Y_i) X_i \mathbb{I}_{r,i}||_2^2$$
$$\leq \sum_{r=1}^{m} ||\frac{1}{\sqrt{m}} A_r (f(W_{t-1}, X_i) - Y_i) \mathbb{I}_{r,i}||_2^2$$
$$= \frac{1}{m} \sum_{r=1}^{m} \mathbb{I}_{r,i} ||(f(W_{t-1}, X_i) - Y_i)||_2^2$$
$$= \frac{1}{m} \sum_{r=1}^{m} 2\mathbb{I}_{r,i} \ell(W_{t-1}, Z_i). \quad (24)$$

In addition, we notice that $\text{Tr}(H_{W_T}(Z)) = \frac{1}{m} \sum_{r=1}^{m} \mathbb{I}_{r,i,T}$. Plugging everything into Lemma 7, we have

$$\text{gen}(\mu, P_{W_T|S}) \leq 3\left(\sum_{t=1}^{T} \frac{R^2 \lambda_t^2 T}{4nm} \mathbb{E}_{W_{t-1}, Z_i}\left[\sum_{r=1}^{m} \mathbb{I}_{r,i,t} \ell(W_{t-1}, Z_i)\right] \mathbb{E}_{W_T, Z_i}\left[\frac{1}{m}\sum_{r=1}^{m} \mathbb{I}_{r,i,T}\right]\right)^{\frac{1}{3}}.$$

This completes the proof. $\qquad\square$

# E    EXPERIMENT DETAILS

## E.1    ARCHITECTURES AND HYPERPARAMETERS

In Section 5, MLP has one hidden layer with 512 hidden units, and AlexNet has five convolution layers (conv. $3 \times 3$ (64 filters) $\rightarrow$ max-pool $3 \times 3 \rightarrow$ conv. $5 \times 5$ (192 filters) $\rightarrow$ max-pool $3 \times 3 \rightarrow$ conv. $3 \times 3$ (384 filters) $\rightarrow$ conv. $3 \times 3$ (256 filters) $\rightarrow$ conv. $3 \times 3$ (256 filters) $\rightarrow$ max-pool $3 \times 3$) followed by two fully connected layers both with 4096 units and a 10-way linear layer as the output layer. All of the convolution layers and the fully connected layers use standard rectified linear activation functions (ReLU).

The fixed learning rates used for MLP and AlexNet are 0.01 and 0.001, respectively. The batch size is set to 60. For the corrupted label experiment, we train the models until the models achieve 100% training accuracy. For other cases, we train the neural networks until the training loss converges (e.g., < 0.0001). Other settings are either described in Section 5 or apparent in the figures. Standard techniques such as weight decay and batch normalization are not used.

To choose the variance proxy $R$, we first collected all the per-instance losses $\ell(W_{t-1}, Z_i)$ that were observed during training, then we let $R = (max_{i,t}\ell(W_{t-1}, Z_i) - min_{i,t}\ell(W_{t-1}, Z_i))/2$ in our experiments.

In Section 6, we compare GMP with other advanced regularization methods. The results of other methods are reported directly from Zheng et al. (2021), and we now give their hyperparameter settings here for completeness. For Dropout, 10% of neurons are randomly selected to be deactivated in each layer. For label smoothing, the coefficient is 0.2. For flooding, the level is set to 0.02. For MixUp, we lineally combine random pairs of training data where the coefficient is drawn from $Beta(1, 1)$. For adversarial training, the perturbation size is 1 for each pixel and we take one step to generate adversarial examples. For AMP, the number of inner iteration is 1, and the $L_2$ norm ball radius values are 0.5 for PreActResNet18 and 0.1 for VGG16, respectively.

The implementation in this paper is on PyTorch, and all the experiments are carried out on NVIDIA Tesla V100 GPUs (32 GB).

## E.2    MORE EXPERIMENTS ON THE COMPARISON BETWEEN THEOREM 2 AND LEMMA 2

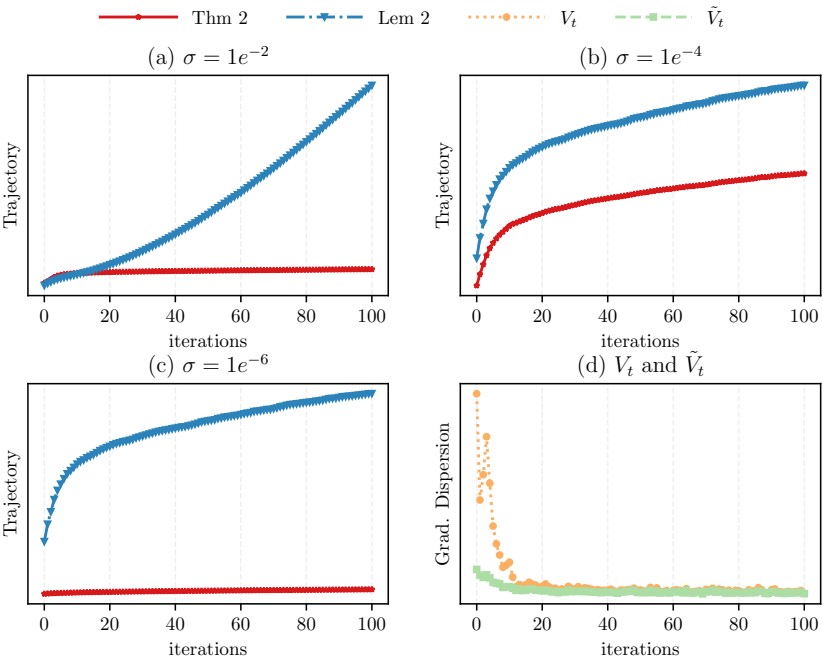

Figure 5: Comparison Between Theorem 2 and Lemma 2 in stochastic setting.

In Figure 1, we compare $\sqrt{\frac{2}{n}\sum_{t=1}^{T}\mathbb{E}\left[\Psi(W_{t-1})+\widetilde{\mathbb{V}}_t(W_{t-1})\right]}$ and $\sqrt{\frac{1}{n}\sum_{t=1}^{T}\mathbb{E}\left[\mathbb{V}_t(W_{t-1})\right]}$ for two different values of $\sigma$. Notice that we indeed use a weaker version of the trajectory term in Theorem 2, and the same constants like $R$, $\lambda_t$ and $\sigma_t$ in the two bounds are ignored here. We choose the full dataset of MNIST and CIFAR10 to train the models with the fixed initialization. We also fix the sampling of batches, making the training completely deterministic. To estimate the local gradient sensitivity term, we randomly sample 20 Gaussian noises from $\mathcal{N}(0,\sum_{\tau=1}^{t-1}\sigma_\tau)$ to perturb the model parameters and compute the average perturbed gradient. With this single draw and deterministic setting, the estimated $\widetilde{\mathbb{V}}_t$ and $\mathbb{V}_t$ are the same, so the bound of Eq. 2 in Theorem 2 should be smaller than the bound in Lemma 2, as shown in Figure 1.

We also provide the experiments under the stochastic setting. Specially, We randomly choose $1/10$ of the MNIST data and train the MLP model with a fixed learning rate and batch size. To estimate $\underset{W_{t-1},Z}{\mathbb{E}}[\nabla\ell(W_{t-1},Z)]$, we conduct 20 runs with different random seeds and save $W$ after every iteration. In Figure 5 (d), we can see that at the beginning of training, our gradient dispersion $\mathbb{V}_t$ is much larger than $\widetilde{\mathbb{V}}_t$ in Neu et al. (2021), but in the later training phase, the magnitude of gap between these two terms is very small. This is because the gradient norm will become tiny when $W$ is near local minima. In Figure 5 (a-c), we compare $\sqrt{\frac{4}{n}\sum_{t=1}^{T}\frac{\lambda_t^2}{\sigma_t^2}\mathbb{E}\left[\Psi(W_{t-1})+\widetilde{\mathbb{V}}_t(W_{t-1})\right]}$ and $\sqrt{\frac{d}{n}\sum_{t=1}^{T}\log\left(\frac{\lambda_t^2\mathbb{E}[\mathbb{V}_t(W_{t-1})]}{d\sigma_t^2}+1\right)}$ under three different settings of $\sigma$. When $\sigma$ is small (e.g., $\sigma=1e^{-6}$), the local gradient sensitivity term will become small, but the factor $1/\sigma^2$ will be very large, making the gap between $\log(x+1)$ and $x$ be extremely large, as shown in Figure 5 (c). In this case, the improvement upon Neu et al. (2021) is significant. When $\sigma$ is large (e.g., $\sigma=1e^{-2}$), at the beginning of training, the trajectory term in Lemma 2 will be smaller than the trajectory term in Theorem 2 since our $\mathbb{V}_t$ is relatively large. However, since $\Psi(W_{t-1})$ has the cumulative variance, and $\mathbb{V}_t$ and $\widetilde{\mathbb{V}}_t$ become closer soon or later, the bound in Lemma 2 will be greater than the bound in Theorem 2 in the later training phase.

### E.3 LEARNING RATE AND BATCH SIZE.

The learning rate and batch size have some impact on Eq. 3 in Theorem 2. We now investigate this by performing experiments with varying learning rates and batch sizes. In our experiments, the model is continuously updated until the average training loss drops below 0.0001. We separate trajectory and flatness terms of the bound and plot them in Figure 6.

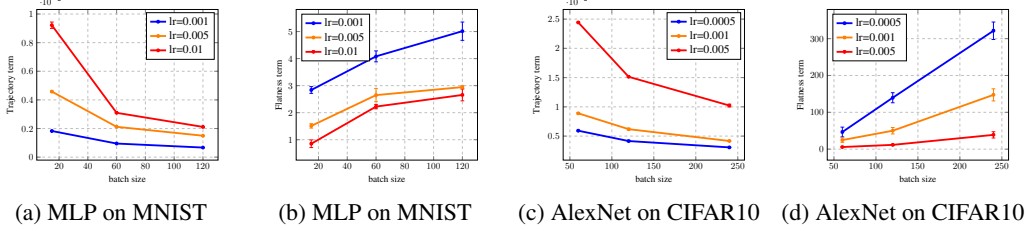

(a) MLP on MNIST    (b) MLP on MNIST    (c) AlexNet on CIFAR10    (d) AlexNet on CIFAR10

Figure 6: The impact of learning rate and batch size on the trajectory term and the flatness term in Eq. 3

A key observation in Figure 6 is that the learning rate impacts the trajectory term and the flatness term in opposite ways, as seen, for example, in (a) and (b), where the two set of curves swap their orders in the two figures. On the other hand, the batch size also impacts the two terms in opposite ways, as seen in (a) and (b) where curves decrease in (a) but increase in (b). This makes the generalization bound, i.e., the sum of the two terms, have a rather complex relationship with the settings of learning rate and batch size. This relationship is further complicated by the fact that a small learning rate requires a longer training time, or a larger number $T$ of training iterations, which increases the number that are summed over in the trajectory term. Nonetheless, we do observe that a smaller batch size gives a lower value of the flatness term ((b) and (d)), confirming the previous wisdom that small batch sizes enable the neural network to find a flat minima (Keskar et al., 2017).

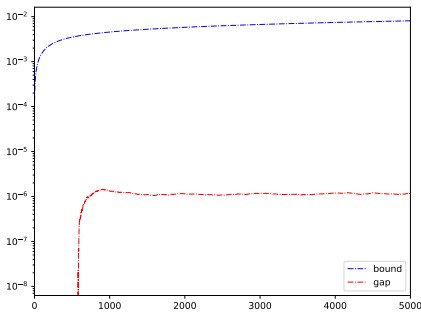

Figure 7: Generalization gap of a linear network v.s. Theorem 3. Note y-axe is log-scale.

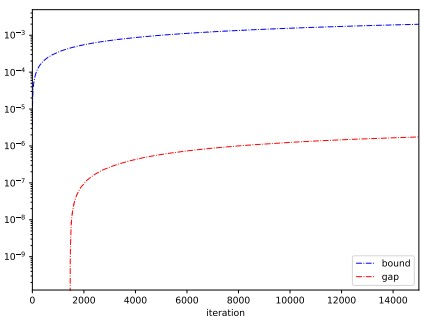

Figure 8: Generalization gap of a two-layer ReLU network v.s. Theorem 4.

### E.4 IMPLEMENTATION OF THEOREM 3 AND THEOREM 4

We let $d_0 = 200$ and use a two-layer ReLU network with hidden units $10000$ to generate $Y$. Moreover, we apply a $\tanh$ function to the output of this network so that $|Y| \leq 1$. The input data $X \sim \mathcal{N}(0, I)$ and we normalize $X$ before training so that $||X|| = 1$. For the training phase, we choose $m = 100$, let data size be $20,000$, batch size be $100$ and learning rate be $0.5$. Figure 7 and Figure 8 compare the empirical generalization gap with the bound in Theorem 3 and that in Theorem 4, respectively.

### E.5 ALGORITHM OF DYNAMIC GRADIENT CLIPPING AND ADDITIONAL RESULTS

The dynamic gradient clipping algorithm is described in Algorithm 1. For both MLP and AlexNet, we let $\alpha = 0.1$. The start step for clipping, $T_c$, is also an important hyperparameter. However, it can be removed by detecting the evolution of the average gradient norm for each epoch. Specifically, whenever the average gradient norm of epoch $j$ is larger than the average gradient norm of epoch $j - 1$, the clipping operation begins.

From Figure 4 and Figure 9 we can see that dynamic gradient clipping effectively alleviates overfitting by conspicuously slowing down the transition of training to the memorization regime, without changing the convergence speed of testing accuracy. Unfortunately, the current design of the dynamic gradient clipping algorithm does not provide a significant improvement for models trained on a clean dataset (without label noise). Designing better regularization algorithms may require understanding the epoch-wise double descent curve of gradient dispersion where the model is trained on a clean dataset.

### E.6 DISCUSSION ON GRADIENT DISPERSION OF MODELS TRAINED ON CLEAN DATASETS

In the case of no noise injected, Figure 3a shows that the model with good generalization property has a exponentially-decaying gradient dispersion. This is consistent with our discussion of Lemma 5 in Section 3, that is, small $I(G_t + N_t; S|\widetilde{W}_{t-1})$ indicates good generalization. Notably, gradient dispersion of AlexNet trained on the real CIFAR10 data still has a epoch-wise double descent curve.

---

**Algorithm 1** Dynamic Gradient Clipping

---

**Require:** Training set $S$, Batch size $b$, Loss function $\ell$, Initial model parameter $\boldsymbol{w}_0$, Learning rate $\lambda$, Initial minimum gradient norm $\mathcal{G}$, Number of iterations $T$, Clipping parameter $\alpha$, Clipping step $T_c$

1: **for** $t \leftarrow 1$ to $T$ **do**
2:    Sample $\mathcal{B} = \{\boldsymbol{z}_i\}_{i=1}^b$ from training set $S$
3:    Compute gradient:
        $g_{\mathcal{B}} \leftarrow \sum_{i=1}^b \nabla_{\boldsymbol{w}} \ell(\boldsymbol{w}_{t-1}, \boldsymbol{z}_i)/b$
4:    **if** $t > T_c$ **then**
5:      **if** $\|g_{\mathcal{B}}\|_2 > \mathcal{G}$ **then**
6:        $g_{\mathcal{B}} \leftarrow \alpha \cdot \mathcal{G} \cdot g_{\mathcal{B}}/\|g_{\mathcal{B}}\|_2$
7:      **else**
8:        $\mathcal{G} \leftarrow \|g_{\mathcal{B}}\|_2$
9:      **end if**
10:   **end if**
11:   Update parameter: $\boldsymbol{w}_t \leftarrow \boldsymbol{w}_{t-1} - \lambda \cdot g_{\mathcal{B}}$
12: **end for**

---

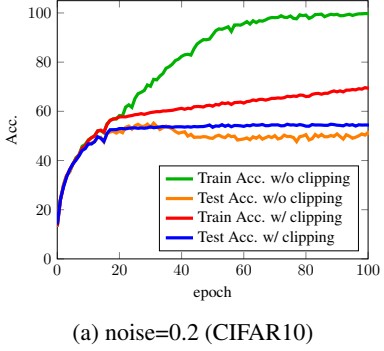

(a) noise=0.2 (CIFAR10)

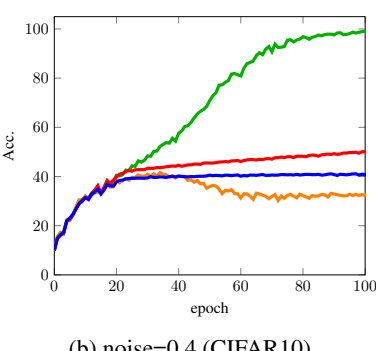

(b) noise=0.4 (CIFAR10)

Figure 9: Dynamic Gradient Clipping (AlexNet).

The difference between Figure 3e with Figure 3f-3h is that the testing accuracy does not decrease in the second phase/memorization regime for AlexNet trained on the true CIFAR10 data. Loosely speaking, we conjugate that memorizing random labels will hurt the performance on unseen clean data but memorizing clean (or true) labels will not. This may explain why dynamic gradient clipping or preventing the training entering the memorization regime cannot improve the performance on a clean dataset.

### E.7    ALGORITHM OF GAUSSIAN MODEL PERTURBATION AND ADDITIONAL RESULTS

---

**Algorithm 2** Gaussian Model Perturbation Training

---

**Require:** Training set $S$, Batch size $b$, Loss function $\ell$, Initial model parameter $\boldsymbol{w}_0$, Learning rate $\lambda$, Number of noise $k$, Standard deviation of Gaussian distribution $\sigma$, Lagrange multiplier $\rho$
    **while** $\boldsymbol{w}_t$ not converged **do**
2:    Update iteration: $t \leftarrow t + 1$
       Sample $\mathcal{B} = \{\boldsymbol{z}_i\}_{i=1}^b$ from training set $S$
4:    Sample $\Delta_j \sim \mathcal{N}(0, \sigma^2)$ for $j \in [k]$
       Compute gradient:
        $g_{\mathcal{B}} \leftarrow \sum_{i=1}^b \left( \nabla_{\boldsymbol{w}} \ell(\boldsymbol{w}_t, z_i) + \rho \sum_{j=1}^k \left( \nabla_{\boldsymbol{w}} \ell(\boldsymbol{w}_t + \Delta_j, z_i) - \nabla_{\boldsymbol{w}} \ell(\boldsymbol{w}_t, z_i) \right)/k \right)/b$
6:    Update parameter: $\boldsymbol{w}_{t+1} \leftarrow \boldsymbol{w}_t - \lambda \cdot g_{\mathcal{B}}$
    **end while**

---

| Method | SVHN | CIFAR-10 | CIFAR-100 |
|---|---|---|---|
| ERM | 97.05±0.063 | 94.98±0.212 | 75.69±0.303 |
| Dropout | 97.20±0.065 | 95.14±0.148 | 75.52±0.351 |
| L.S. | 97.22±0.087 | 95.15±0.115 | 77.93±0.256 |
| Flooding | 97.16±0.047 | 95.03±0.082 | 75.50±0.234 |
| MixUp | 97.26±0.044 | 95.91±0.117 | 78.22±0.210 |
| Adv. Tr. | 97.23±0.080 | 95.01±0.085 | 74.77±0.229 |
| AMP | **97.70±0.025** | **96.03±0.091** | **78.49±0.308** |
| **GMP**$^3$ | 97.43±0.037 | 95.64±0.053 | 78.05±0.208 |
| **GMP**$^{10}$ | 97.34±0.058 | 95.71±0.073 | 78.07±0.170 |

Table 2: Top-1 classification accuracy acc.(%) of PreActResNet18. We run experiments 10 times and report the mean and the standard deviation of the testing accuracy. Superscript denotes the number of sampled Gaussian noises during training.

| Method | SVHN | CIFAR-10 | CIFAR-100 |
|---|---|---|---|
| **GMP**$^3_{\text{abs}}$ | 97.10±0.054 | 94.21±0.139 | 74.80±0.113 |

Table 3: Top-1 classification accuracy acc.(%) of VGG16.

The GMP algorithm is given in Algorithm 2. Table 1 and Table 2 show that our method is competitive to the state-of-the-art regularization techniques. Specifically, our method achieves the best performance on SVHN for both models and on CIFAR-100 where VGG16 is employed. Particularly, testing accuracy is improved by nealy 2% on the CIFAR-100 dataset with VGG16. For other tasks, GMP is always able to achieve the top-3 performance. In addition, we find that increasing the number of sampled noises does not guarantee the improvement of testing accuracy and may even degrade the performance on some datasets (e.g., SVHN). This hints that we can use small number of noises to reduce the running time without losing performance. Moreover, we observe that **GMP with $k = 3$ usually takes around** $1.76\times$ **that of ERM training time**, which is affordable.

For PreActResNet18, the performance of GMP appears slighly inferior to the most recent record given by AMP (Zheng et al., 2021). Noting that the key ingredient of AMP, "max-pooling" in the parameter space, greatly resembles regularization term in GMP, which may be seen as "average-pooling" in the same space.

One potential extension of GMP is to let the variance of the noise distribution be a function of the iteration step $t$. In other words, using the time-dependent $\sigma_t$ instead of a constant $\sigma$.

We also consider the following regularized scheme, which is a absolute value version.

$$\min_w L_s(w) + \rho \underset{\Delta \sim \mathcal{N}(0, \sigma^2 \mathbf{I}_d)}{\mathbb{E}} \left[ \left| L_s(w + \Delta) - L_s(w) \right| \right].$$

This scheme can still perform well as shown in Table 3. In fact, it outperforms GMP$^3$ on CIFAR-100. This hints that it is possible to improve the performance by choosing other norm of $L_s(w + \Delta) - L_s(w)$.

