# OpenReview forum: "On the Generalization of Models Trained with SGD: Information-Theoretic Bounds and Implications"
_ICLR.cc/2022/Conference — ICLR 2022 Poster_

### Official Review · Reviewer_A9tc · 2021-10-30

**Correctness:** 4
**Technical Novelty And Significance:** 4
**Empirical Novelty And Significance:** 3
**Recommendation:** 10
**Confidence:** 3

**Main Review:**

The paper is exceptionally well written. The contribution is important not only from a theoretical, but also from a practical perspective, and the insights and hypotheses derived have the potential to bring the field a significant step forward. The authors also cover and connect with the related literature, at least as far as I can tell from my limited knowledge of this field. The explanations after Lemma 5 and Theorem 2 are particularly nice, and so is the discussion of Dynamic Gradient Clipping.

I do have some questions that I would appreciate to have addressed during the discussion period, but these are mostly out of my own curiosity:

- First, from Th. 3 and 4 it becomes clear that the generalization error increases with the number of epochs. This is also seen in Fig. 1. I assume that this is natural for the first few epochs, as initially the model performs equally badly on the training data and on the population. Is this the correct interpretation? However, what is the intuition behind a monotonic increase of the generalization error over epochs? Especially in the linear model, the term |W_t-1|^2 will eventually settle to a constant (if we can assume that training is successful). Does this mean that essentially the bound becomes more and more loose as T increases? Figure 8 suggests so.
- In Figure 7, it seems that the trajectory term and flatness term have different orders of magnitude. This somehow works against your explanation that the two terms behave in opposite ways when it comes to batch size and learning rate. The picture is rather clear when you consider the orders of magnitude.
- Does the second term in (1) really measure flatness? Suppose that w_1 is a very deep local minima, such that \gamma evaluated at s is large for many samples s (e.g., if s is a large enough sample). In this case, we may assume that the flatness term is small, since both \gamma values (at S and S') cancel, despite the fact that w_1 is not flat. What am I missing?
- In Th. 3 and 4, is it still possible to dissect the bound into the product of a Hessian and a gradient dispersion term? If so, please mention that in the text.
- In respect to Figure 4, why is gradient dispersion small at later epochs? I would assume that at a minimum of w, the gradient direction fluctuates more strongly with different samples than it does during initial training (where a strong gradient signal is available). Is this related to the flatness of the minimum, where the gradient magnitude is small for almost all Z? Does, in this sense, also the trajectory term contain some information about flatness?
- In your regularization scheme, why do you not propose regularizing with the magnitude (or some other norm) of L_s(w+\Delta)-L_s(w)?
- How did you obtain the standard deviations (?) in Table 1? How many independent instances did you train?
Furthermore, I have a few editorial comments:
- In some instances, the text could do with a revision (e.g., p1: "in fact broadly apply broadly", p2: "is presented, which the bound is revised", p2: "We also provide an application of our bounds is in analyzing", etc.).
- The distinction between sample-level and instance-level mutual information was not clear to me in the introduction, even though I am familiar with both types of bounds. Have these terms become common?
- The usage of the batch gradient function g is not consistent; e.g., in the definition of \psi, g is evaluated at a single sample Z (if I understood it correctly) and thus coincides with \nabla_w l(w,Z); however, in the definition of V(w) the nabla version is used instead of g.
- The HWI of the HWI inequality could be spelled out completely at least once. I am not familiar with this inequality, and I think it would be good to have a name to it.
- Fig. 2 is too small to read.
- In Fig. 3, I suggest to use the same axes for both the gap and the bound. Even though the results may not look as impressive, it would be good to show a fair comparison.


I nevertheless have a few

**Summary Of The Paper:**

The paper presents a new bound on the generalization error of models trained with stochastic gradient descent. The new bound builds on the work of Neu, but is tighter. Based on the insights obtained from the bounds, the authors propose two approaches to improve generalization. On the one hand, they show that gradient clipping helps preventing overfitting. On the other hand, the authors propose a regularization term that adds Gaussian noise to the network weights (Gaussian model perturbation) and show that this type of regularization performs on par with or even better than other contemporary regularization schemes. Aside from these technical contributions, the authors provide ample insight into the mathematical terms they derive and discuss several phenomena observed during learning. In the outlook, the authors discuss how their bound can be further improved by utilizing the strong data processing inequality.

**Summary Of The Review:**

An excellent paper of high theoretical and practical relevance. Very well written, with many interesting insights.

---

> ### Author Response · Authors · 2021-11-23
> **To Reviewer A9tc:**
>
> Thank you very much for your positive comments. Below we discuss your concerns in detail.
>
> >- First, from Th. 3 and 4 it becomes clear that the generalization error increases with the number of epochs. This is also seen in Fig. 1. I assume that this is natural for the first few epochs, as initially the model performs equally badly on the training data and on the population. Is this the correct interpretation?
>
> **Response.**  Yes, your interpretation about the early stage of training is correct.
>
> >- However, what is the intuition behind a monotonic increase of the generalization error over epochs? Especially in the linear model, the term |W_t-1|^2 will eventually settle to a constant (if we can assume that training is successful). Does this mean that essentially the bound becomes more and more loose as T increases? Figure 8 suggests so.
>
>  This is a crucial question of the bounds in our original submission. After fixing the bug in the proof of our previous main theorem, we re-derive these bounds and remove the norm of the model parameters ($||W||_2$) from the bound in our revision. Now as long as the training is successful (i.e., the empirical risk is nearly zero), the bound will not increase anymore. If the empirical risk does not converge to zero during the training, you question is still valid, i.e., the bound becomes looser as T increases, which is inconsistent with our main theorem (Theorem 1). One possible explanation is that the bounds of the linear model and the ReLu network (Theorem 2 and 3) are looser than the bound in our main theorem, due to the additional applications of Cauchy-Schwartz inequality in establishing Theorem 2 and 3.
>
>
> >- In Figure 7, it seems that the trajectory term and flatness term have different orders of magnitude. This somehow works against your explanation that the two terms behave in opposite ways when it comes to batch size and learning rate. The picture is rather clear when you consider the orders of magnitude.
>
> **Response.**  Our previous statement might not be clear. We need to emphasize that our discussion is based on the impact of batch size and learning rate on the Eq.(2), which is the product of the trajectory term and the hessian term (and the noise disappears in the bound), rather than Eq. (1) that is the summation of two terms (and the noise will influent the magnitude of the two terms). Notice that the condition of reaching the optimal bound requires the two terms in Eq. (1) be equal. Thus, although hessian term has larger order, we cannot say the hessian term dominate the change of the bound of Eq (2). What really matters here is the changing rate, rather than the changing magnitude.
>
>
> >- Does the second term in (1) really measure flatness? Suppose that w_1 is a very deep local minima, such that \gamma evaluated at s is large for many samples s (e.g., if s is a large enough sample). In this case, we may assume that the flatness term is small, since both \gamma values (at S and S') cancel, despite the fact that w_1 is not flat. What am I missing?
>
> **Response.**  We believe that your point here is valid, and we have no concrete evidence to exclude the scenario you point to. The name "flatness term" might have exaggerated the role of this term in quantifying flatness. **In the revision, we have noted that this term is correlated with flatness rather than precisely measures the flatness. Specifically, we note that our theorem does not assert that sharp minima will not generalize well. ** On the other hand, to argue that the scenario you point to may not occur often, we note that the two $\gamma$ terms may not cancel each other since even a small shift between $S'$ and $S$ will lead to a large generalization gap if a very sharp minimum is found (see a Figure 1. (a) in [E]). In addition, given a specific $W=w$, two $\gamma$ terms are evaluated under the distributions $P_{S|W=w}$ and $P_{S'}$ respectively and the difference between the two distributions is perhaps unlikely to result in the two $\gamma$ terms cancel each other.
>
> >- In Th. 3 and 4, is it still possible to dissect the bound into the product of a Hessian and a gradient dispersion term? If so, please mention that in the text.
>
> **Response.**  These bounds are already in the form of the product of a hessian term and trajectory term. **In the revision, we specifically make this remark in the text before presenting the results**
>
>
>
> [E] He, Haowei, Gao Huang, and Yang Yuan. "Asymmetric valleys: beyond sharp and flat local minima." Proceedings of the 33rd International Conference on Neural Information Processing Systems. 2019.

---

> > ### Author Response · Authors · 2021-11-23
> > **To Reviewer A9tc (cont.):**
> >
> > >- In respect to Figure 4, why is gradient dispersion small at later epochs? I would assume that at a minimum of w, the gradient direction fluctuates more strongly with different samples than it does during initial training (where a strong gradient signal is available). Is this related to the flatness of the minimum, where the gradient magnitude is small for almost all Z? Does, in this sense, also the trajectory term contain some information about flatness?
> >
> > **Response.**  Thanks for this question. Your reasoning here is in line with our understanding. When training near convergence, the gradient direction strongly fluctuates but maintains a small magnitude due to the flatness of the approached minimum (which is also observed by the work of [F]). This causes the gradient dispersion to be small near convergence. It is interesting that you bring up the question whether the trajectory term may also contain information about flatness. We tend to agree with this statement; but we do not see a clear way to theoretically measure the influence of the SGD trajectory on the flatness of the found minima.
> >
> >
> > >- In your regularization scheme, why do you not propose regularizing with the magnitude (or some other norm) of L_s(w+\Delta)-L_s(w)?
> > How did you obtain the standard deviations (?) in Table 1? How many independent instances did you train?
> >
> > **Response.**  After reading your comments, we conducted some preliminary experiments using **[L1 norm](/wUPmnxbPT9KklbiRcgW6jg)** of $L_s(w+\Delta)-L_s(w)$ as a regularizer. This scheme does offer some improvement over the standard ERM training, but appears to perform inferior to our GMP scheme in most cases. Note that the design of GMP justified by the flatness term in Theorem 1, but such norm-based regularization may not have a strong theoretical basis (that is supported in this work).
> >
> > To obtain Table 1, we run experiments 10 times and report the mean and the standard deviation of the testing accuracy. These details are now added to the title of Table 1 in our revision.
> >
> > >- In some instances, the text could do with a revision (e.g., p1: "in fact broadly apply broadly", p2: "is presented, which the bound is revised", p2: "We also provide an application of our bounds is in analyzing", etc.).
> >
> > **Response.**  Thanks for your suggestions, we have fixed them in the revision.
> >
> > >- The distinction between sample-level and instance-level mutual information was not clear to me in the introduction, even though I am familiar with both types of bounds. Have these terms become common?
> >
> > **Response.**  No, these terms have not become standard, and our choice of such names is just for distinguishing the two. In the revision, we have removed the instance-level mutual information bound since it in fact gives the identical bound.
> >
> > >- The usage of the batch gradient function g is not consistent; e.g., in the definition of \psi, g is evaluated at a single sample Z (if I understood it correctly) and thus coincides with \nabla_w l(w,Z); however, in the definition of V(w) the nabla version is used instead of g.
> >
> >
> > >- Fig. 2 is too small to read.
> >
> >
> > **Response.**  Thanks for pointing these out. We have modified them in the revision.
> >
> > >- The HWI of the HWI inequality could be spelled out completely at least once. I am not familiar with this inequality, and I think it would be good to have a name to it.
> >
> > **Response.** Th strong version of the HWI inequality is relating the relative entropy ($H$), the quadratic transport cost or Wasserstein distance ($W$) and the Fisher information ($I$), so it seems that this so-called HWI does not have a complete spelling form.
> >
> >
> > >- In Fig. 3, I suggest to use the same axes for both the gap and the bound. Even though the results may not look as impressive, it would be good to show a fair comparison.
> >
> > **Response.** We let the gap and the bound in the same y-axis (in log scale) now in the revision.
> >
> > [F] Feng, Yu, and Yuhai Tu. "Phases of learning dynamics in artificial neural networks: in the absence or presence of mislabeled data." Machine Learning: Science and Technology (2021).

---

### Official Review · Reviewer_Xt5A · 2021-10-30

**Correctness:** 3
**Technical Novelty And Significance:** 3
**Empirical Novelty And Significance:** 2
**Recommendation:** 6
**Confidence:** 3

**Main Review:**

This paper is based on (Neu 2021). The authors clearly presented the technical increment, and the empirical improvements. A strength in this paper is that the proposed generalization bound yields a simple practical regularization technique, which can improve classification with VGG16. This is some novel bits as compared to (Neu 2021), which is mostly theoretical.

Despite a few weakness (see below, for the authors to work on a revision), I tend to vote for acceptance, subject to the authors' rebuttal and the reviewers' discussion.

First, this paper largely requires the settings of a very recent work. However, the significance of (Neu, 2021) is not clearly written. Why this
recently published results are such a foundational work to build upon? It is meaningful to compare against another, more well-known, bound. For example, in the figure 1, there can be a third bound, comparing with the sum of the trajectory term and the flatness term.

"let $\xi$ govern the randomness in the sequence": this paragraph explain a main assumption of the results, and is informal. There are two factors: the way that the samples are split, and the way the batches are ordered. It not clear on what is random and what is fixed.

The discussion after theorem 1. There is an inconsistency here. Do $\Psi(W_t)$ in Lemma 3 and $\Phi(W_t)$ here refer to the same thing?
The bounds in Lemma 3 and Theorem 1 should be explained in more detail. As the meaning of those two terms, and which one is the trajectory/flatness term.

Overall, the writing can be better polished to explain the background and provide intuitions.

**Minor comments:**

Introduction: "Neu (2021) presents an information- theoretic analysis" extra space before theoretic

P2
"The bounds we obtain decompose into two terms"
-> "The bounds we obtained can be decomposed into two terms"

Theorem 4: the equation is too wide

Table 1: add some margin between the table and the surrounding text

P3: is the KL divergence or cross entropy sub-Gaussian?

Lemma 3: add a remark to compare against lemma 1 and lemma 2

Statements in conclusion. This is not so meaningful without sufficient motivation and discussion.



**Summary Of The Paper:**

This paper proposed a new generalization bound which is based on the recent construction by Neu (2021). The authors made further assumptions on the randomness of the batch sequence, and improved the results by Neu by achieving tighter bounds by tuning the trajectory term. It is verified empirically, and the flatness term gives a regularization technique for better classification on VGG16 on MNIST.


**Summary Of The Review:**

Pro:
- Novel increments over a recent work on generalization bound
- New regularization based on the flatness term

Con:
- A few glitches in writing

---

> ### Author Response · Authors · 2021-11-23
> **To Reviewer Xt5A:**
>
>   Thank you very much for your careful reading and constructive comments. Our responses follow.
>
>
> >- First, this paper largely requires the settings of a very recent work. However, the significance of (Neu, 2021) is not clearly written. Why this recently published results are such a foundational work to build upon?
>
>
> **Response.** Many researchers think the peculiar generalization behavior of deep neural networks is highly correlated to some properties of SGD and/or the structure of the neural networks. Hence the algorithm-dependent generalization bounds have attracted significant research attention recently. Information-theoretic bounds form a family of the algorithm-dependent generalization bounds and has been successfully appied to SGLD in some recent works and demonstrated a great power in analyzing iterative learning algorithms. Neu'21 is the first work that uses information-theoretic bounds to analyze SGD, and we find the result is intriguing based on two reasons. First, unlike many other generalization bounds that based on the norm of the final model parameters, the information-theoretic bound demonstrates that the whole training trajectory of SGD has explicit effect on the final generalization behavior of the model. This result suggests the importance of studying the generalization of DNN by inspecting the entire optimization process.  In addition, it also justifies that flat minima can lead to better generalization, which is not characterized in some other generalization bounds such as [A,B,C,D]. Accepted in the prestigious learning theory conference COLT 2021, Neu'21 is believed to point to promising directions to further understanding generalization.
>
> >- It is meaningful to compare against another, more well-known, bound. For example, in the figure 1, there can be a third bound, comparing with the sum of the trajectory term and the flatness term.
>
> **Response.**  We agree that it is necessary to compare information theoretic bounds with other generalization bounds. However, it is not easy to compare the current version of our bounds with other type of generalization bounds. Specifically, many existing bounds are high probability bounds while our bounds characterize the expected generalization gap. Furthermore, most other bounds are based on different assumptions. For example, our bound requires that the loss is subGaussian while some other bounds need that the loss has convex, Lipschitz, and smooth conditions, or require that the model is in the overparameterization setting (i.e., let the width of DNN be large enough). If the reviewer is aware of any other bounds, against which our bound can be fairly compared, we will be glad to include such a comparison in the next revision of this paper.
>
> >- "let $\xi$ govern the randomness in the sequence": this paragraph explain a main assumption of the results, and is informal. There are two factors: the way that the samples are split, and the way the batches are ordered. It not clear on what is random and what is fixed.
>
> **Response.**  Here $\xi$ governs both the random shuffling of the training set and the random split of it into an ordered sequence of batches.  When assuming $\xi$ is fixed, we consider both sources of randomness are fixed. This treatment solely serves the purpose of lightening the expression. There is no technical difficulty to include such randomness in our development.
>
>
> >- The discussion after theorem 1. There is an inconsistency here. Do $\psi$ in Lemma 3 and $\phi$ here refer to the same thing? The bounds in Lemma 3 and Theorem 1 should be explained in more detail. As the meaning of those two terms, and which one is the trajectory/flatness term.
>
> **Response.** Yes, they refer to the same thing. Thanks for spotting this typo. Section 3 has been re-organized in the revision, please let us know if you have additional comments.
>
> [A] Hardt, Moritz, Ben Recht, and Yoram Singer. "Train faster, generalize better: Stability of stochastic gradient descent." ICML 2016.
>
> [B] Bartlett, Peter L., Dylan J. Foster, and Matus J. Telgarsky. "Spectrally-normalized margin bounds for neural networks." NeurIPS 2017.
>
> [C]Arora, Sanjeev, et al. "Fine-grained analysis of optimization and generalization for overparameterized two-layer neural networks." ICML 2019.
>
> [D] Cao, Yuan, and Quanquan Gu. "Generalization bounds of stochastic gradient descent for wide and deep neural networks." NeurIPS 2019.

---

> > ### Author Response · Authors · 2021-11-23
> > **To Reviewer Xt5A (cont.):**
> >
> > >- Minor comments:
> > >Introduction: "Neu (2021) presents an information- theoretic analysis" extra space before theoretic
> > >P2 "The bounds we obtain decompose into two terms" -> "The bounds we obtained can be decomposed into two terms"
> > >Theorem 4: the equation is too wide
> > >Table 1: add some margin between the table and the surrounding text
> >
> >
> > **Response.** Thanks for providing these suggestions, we have fixed all of them in the revision.
> >
> > >- P3: is the KL divergence or cross entropy sub-Gaussian?
> >
> > **Response.**
> > During SGD training, cross-entropy loss values encountered arguably follow a subGaussian distribution since they remain bounded from above. We however did not attempt a rigorous proof for such a claim, since the proof is likely to bring more non-essential technicality into the paper and merely serves as a digression from the main theme of this work.
> >
> > >Statements in conclusion. This is not so meaningful without sufficient motivation and discussion.
> >
> > **Response.**   The discussion of strong data processing inequality in the conclusion is now removed.

---

### Official Review · Reviewer_ULcg · 2021-10-31

**Correctness:** 4
**Technical Novelty And Significance:** 3
**Empirical Novelty And Significance:** 3
**Recommendation:** 5
**Confidence:** 3

**Main Review:**

This paper is solid and well-structured. It has a very clear introduction part, making readers easy to follow their idea. On the other hand, compared to Neu et. al., the bound derived in this paper is much tighter (see figure 1), and the technique to derive this is very interesting. However, there do exist some small issues, which is listed as below:
1. The main concern comes from the novelty. The author also admitted that their bounds are not new compared with the previous result. On the other hand, their application for linear model, and two layer ReLU network is not convincing enough. For example, they author claims that the bound might be independent with the model complexity in the generalization bound due to the fact that the deactivated neurons do not contribute to the bound. However, it is not clear that how many neurons are activated from this bound, which means that this bound itself is not useful. On the other hand, the authors argue that their bound decays with the increase of the model size. This is not fair since they only run the simulation for different widths and filters (Figure 3a, b). It would be great if they can provide experiments for the depth.
2. Some arguments are provided without any explanation. For example, in page 5, the last second paragraph of Theorem 2, the author mentioned that “This justifies the construction of the auxiliary weight process”. It is a little confused for me that based on the previous description, without the auxiliary weight process the bound will include an additional $log(n)$ factor, which is not a big problem to me. Another example, in the Experimental Study section, author said that “replace the labels of ε fraction of the training and testing instances with random labels”. Is the label in testing instances also replaced by the random labels?  Moreover, in the Dynamic Gradient Clipping section, author said that “i.e., the model is expected to have entered the “memorization” regime”. It is quite confusing for me that why the gradient norm is large than the gradient norm K steps earlier can lead to this argument.

**Summary Of The Paper:**

This paper derives an improved stability-based generalization bound for SGD upon Neu et. al.’s work. In particular, they drop a term called local gradient sensitivity, leading to significantly tighter bound in practical models. Based on this generalization bound, they study both linear model and two layer ReLU neural networks. During their analysis, they obtain several interesting observations, which is consistent to the previous ones. Finally, based on their bound, they propose a new training scheme, named Gaussian model perturbation, having good performance on popular datasets.

**Summary Of The Review:**

This paper is good and well-structured. However, its novelty is not very strong and there exists some confusing arguments.

---

> ### Author Response · Authors · 2021-11-23
> **To Reviewer ULcg:**
>
>  Thank you for your constructive comments. Our responses follow.
>
> >- The main concern comes from the novelty. The author also admitted that their bounds are not new compared with the previous result.
>
> **Response.** Agreeably our construction of the auxiliary weight process follows that in Neu'21. But this work differs from and improves upon Neu'21 in several ways.
> 1. The new bound presented in this paper removes the local gradient sensitivity term in Neu'21 and is much tighter.
> 2. Our proof technique is different from Neu'21. Specifically, their proof relies on utilizing another ghost auxiliary weight process, giving rise to a pessimistic bound that includes the local gradient sensitivity term. In our proof, the ghost process is completely removed. We now explicitly show this difference in our revision (see Lemma 4).
> 3.  Our bound allows us to obtain a simple closed form expression of the optimal bound with respect to noise variances. This simplifies the computation of the bound, without the need of explicit selecting the noise variance as in Neu'21 and numerically optimizing over the choices of the variance.
>
> We think these differences and improvements adequately justify the novelty of this paper.
>
> >- On the other hand, their application for linear model, and two layer ReLU network is not convincing enough. For example, they author claims that the bound might be independent with the model complexity in the generalization bound due to the fact that the deactivated neurons do not contribute to the bound. However, it is not clear that how many neurons are activated from this bound, which means that this bound itself is not useful.
>
> **Response.**  The insight that generalization can be related to the deactivation states of a ReLU network, to the best of our knowledge, is for the first time revealed from a theoretical angle. We disagree that such a perspective is useless. At the worst, one may utilize this result empirically by tracking the deactivation states of neurons to study generalization (and this is what we exactly did to plot Figure 2). Of course, if a theoretical approach can be developed to track the neuron states would further strengthen the usefulness of this result. But that appears a separate topic in its own right, well beyond the scope of this paper.
>
> >- On the other hand, the authors argue that their bound decays with the increase of the model size. This is not fair since they only run the simulation for different widths and filters (Figure 3a, b). It would be great if they can provide experiments for the depth.
>
> **Response.** Thank you for this suggestion. We also performed some experiments varying the depth of neural networks. Although it seems that the results of depth are not as impressive as the results of width, some similar observations are obtained for MLP, for example, MLP with depth 3 has better performance and smaller estimated bound values than that of MLP with only one hidden layer. It's worth mentioning that unlike experiments on width, the training of networks with different depth requires more careful hype parameters tuning. We will continue to investigate this and add the results in the future revision after conducting more experiments.

---

> > ### Author Response · Authors · 2021-11-23
> > **To Reviewer ULcg (cont.):**
> >
> > >- For example, in page 5, the last second paragraph of Theorem 2, the author mentioned that “This justifies the construction of the auxiliary weight process”. It is a little confused for me that based on the previous description, without the auxiliary weight process the bound will include an additional $log(n)$ factor, which is not a big problem to me.
> >
> > **Response.**  Characterizing how the generalization error decays with $1/n$ or $1/\sqrt{n}$ is a fundamental task in statistical learning theory, in which one seeks to find the tightest rate of convergence. The $\log n$ factor would make the bound much looser, particularly for large $n$.
> >
> >
> > >- Another example, in the Experimental Study section, author said that “replace the labels of ε fraction of the training and testing instances with random labels”. Is the label in testing instances also replaced by the random labels?
> >
> > **Response.** Yes, and this is mentioned in the main text. Both training dataset and testing dataset are replaced by the random labels because we follow the traditional learning theory setting where testing data and training data come from the same distribution. This is also the reason that we say the accuracy of about 80% is a near-optimal testing accuracy in Figure 5(a), where the label noise level of MNIST is 0.2 (for both training and testing).
> > >- Moreover, in the Dynamic Gradient Clipping section, author said that “i.e., the model is expected to have entered the “memorization” regime”. It is quite confusing for me that why the gradient norm is large than the gradient norm K steps earlier can lead to this argument.
> > **Response.** In this work, the division of the memorization regime and the generalization regime is concluded via empirical observation. Due to the lack of a precise definition of the dividing point, we devise this criterion to tolerate our inaccuracy in defining the dividing point.In addition, allowing gradient norm to grow $K$ more steps will accelerate the convergence of the training loss in practice.

---

### Official Review · Reviewer_RzUv · 2021-11-04

**Correctness:** 3
**Technical Novelty And Significance:** 3
**Empirical Novelty And Significance:** 3
**Recommendation:** 8
**Confidence:** 4

**Main Review:**

Two comments:
1.	The introduction of Wasserstein distance in the main body of the paper seems to be redundant, as what is really used in the proof is the eq (3) in the appendix, which is only related to squared error between X and Y. A more concise way to present the result could be put eq (3) in the main body of the paper and discuss its connection to Wasserstein distance and HWI inequality in the appendix.
2.	In evaluating the bounds in Figures 2 and 3, how to compute the sub-Gaussian parameter R? If I understand correctly, the loss function for MLP and AlexNet should be cross-entropy, which is not bounded. Moreover, can you prove that they are subgaussian?

Minor comments:
1.	The organization of section 3 can be improved. Currently, the statement of the main theorem, discussion, sketch of proof, and comparison are stuck together.
2.	Plots in Figure 3 have two different y-axes, which is hard for me to compare the true generalization gap and the bound. Can you somehow calibrate these plots?


**Summary Of The Paper:**

This paper presents novel and tighter information-theoretic upper bounds for the generalization error of machine learning models, such as neural networks, trained with SGD. Following the same construction of the auxiliary weight process in Neu (2021), the upper bounds proposed in this paper improve upon Neu (2021) in two ways. One improvement is mainly due to removing an unnecessary term in the bound of Neu (2021), which the author refers to as “local gradient sensitivity”, by invoking the HWI inequality (Raginsky & Sason, 2018). Additionally, the improvement also benefits from replacing a sample-level mutual information term in Neu (2021) with an individual sample mutual information term, exploiting a recent result of Bu et al. (2020). The bounds obtained here decompose into two terms, one measuring the impact of training trajectories (“the trajectory term”) and the other measuring the impact of the flatness of the found solution (“the flatness term”).

The authors also apply these bounds to analyze the generalization behavior of linear and two-layer ReLU networks. Experimental studies based on these bounds provide some insights into the SGD training of neural networks. Some simple regularization schemes, including dynamical gradient clipping and Gaussian model perturbation, are proposed, which are shown to perform comparably to the current state of the art.


**Summary Of The Review:**

This is a great paper that refines the information-theoretical analysis of the generalization behavior of SGD using the HWI inequality and the ISMI bound.

---

> ### Author Response · Authors · 2021-11-23
> **To Reviewer RzUv:**
>
> We thank you sincerely for providing comments to our paper. Below we discuss your concerns in detail.
>
> >- The introduction of Wasserstein distance in the main body of the paper seems to be redundant, as what is really used in the proof is the eq (3) in the appendix, which is only related to squared error between X and Y. A more concise way to present the result could be put eq (3) in the main body of the paper and discuss its connection to Wasserstein distance and HWI inequality in the appendix.
>
> **Response.** We removed the Wasserstein distance from the main text in our revision and used the squared error between X and Y in the corresponding lemma.
>
>
>
> >- In evaluating the bounds in Figures 2 and 3, how to compute the sub-Gaussian parameter R?
>
> **Response.**
> In our original submission, to choose the variance proxy $R$, we first collected all the per-instance losses that were observed during training, then we used an Gaussian distribution to fit these loss values and we found that the standard deviation of the learned Gaussian is between $0.1\sim0.3$. We thus select $R=0.1$.
>
> It's important to mention that we use the different $R$ in the revision. To be precise, we first collected all the per-instance losses that were observed during training, then we let $R=(max\\_ loss-min\\_ loss)/2$ in our experiments.
>
> We inadvertently omit these details in our original submission, and they have been added in the revision.
>
> >- If I understand correctly, the loss function for MLP and AlexNet should be cross-entropy, which is not bounded. Moreover, can you prove that they are subgaussian?
>
> **Response.** Indeed the cross-entropy loss is used in our training.  Albeit the unboundedness of cross-entropy loss, we believe that it indeed has a subGaussian distribution under the SGD training since the loss remains bounded (and decreasing) during training. We have not made an effort in rigorously proving the subGaussianality of the loss. Such an effort would require a careful examination of the SGD dynamics and involve significant technicality which seems to only serve as a digression from the main thrust of the paper. In paractice, to estimate the variance proxy of the loss when treating it as a subGaussian random variable, one can take the maximum seen value of the loss as its upper bound,  and use this bound to obtain an estimate of the variance proxy. But we believe that such an estimate is already a significant over-estimate.
>
>
> >- The organization of section 3 can be improved. Currently, the statement of the main theorem, discussion, sketch of proof, and comparison are stuck together.
>
> **Response.** Section 3 has been re-organized and become more readable now, please let us know if you have any additional comments on it.
>
>
> >- Plots in Figure 3 have two different y-axes, which is hard for me to compare the true generalization gap and the bound. Can you somehow calibrate these plots?
>
> **Response.** Although we mainly want to show that our bound can characterize the dynamics of the generalization gaps, we agree it is necessary to compare the magnitude of gap value and the magnitude of bound value fairly. We have modified those figures in our revision according to this suggestion.

---

### Author Response · Authors · 2021-11-23
**To all reviewers:**

We would like to thank all reviewers for your insightful comments. We have revised the paper to address these comments and we will discuss these revisions separately in our response to each reviewer.

Additionally, we wish to report that we discover a bug in our original proof of the theorems. We have made an effort fixing this bug, which leads to a revision of the presented theorems. Specifically,

1. The definition of gradient dispersion is revised. Previously, the definition is: $\mathbb{E}_Z[||\nabla_w\ell(w,Z)-\mathbb{E}_Z\mathbb[\nabla_w\ell(w,Z)]||^2_2]$. The current definition is $\mathbb{E}_S[||g(w,B_t)-\mathbb{E}_\{W,Z\}[\nabla_w\ell(W,Z)]||^2_2]$.
2. Theorem 1 (sample-level mutual information bound) is revised according the new definitioin of gradient dispersion. And the optimal version of the bound is also contained.
3. The original Theorem 2 (instance-level mutual information bound) is now removed, since under the new definition of gradient dispersion, the bound coincides with the sample-level mutual information bound in Theorem 1.
4. The previous Theorems 3 and 4 are now presented as Theorems 2 and 3, revised corresponding to the change of gradient dispersion definition. We also rearrange the terms in the bounds and make the form of these two bounds be more compact.


The main result of the paper is about removing local gradient sensitivity. The new version of theorems maintains the key results. In fact, the update proof techniques improve the novelty compared with Neu'21.
Specifically, the development in Neu'21 requires constructing a ghost auxiliary weight process in which an independent noise perturbation $\Delta_t'$ is introduced. Due to the mismatch between $\Delta_t$ and $\Delta_t'$, the local gradient sensitivity term $\Psi(W_{t-1})$ appears in their bound. In our revision, we completely avoid using such a ghost auxiliary weight process by applying Lemma 4 (see our revision), which indeed simplifies the proof. On top of that, exploiting Lemma 4 will also help to omit some applications of triangle inequality in the original proof. This enables us to obtain the smaller constant factor in the bound (e.g., our constant factor is half of that in Neu'21).

We also have made revision in experiments. In a nutshell, main experimental conclusion stays the same. In particular, updated Figure 1 still shows that our bound is better than Neu'21. This result can be view as conveying a message that the key difference between our new version of bounds and Neu'21 is not the difference of definition  in gradient dispersion but is whether relying on a ghost auxiliary weight process or not.

---

### Decision · Program_Chairs · 2022-01-20

**Decision:**

Accept (Poster)

**Comment:**

This paper offers a refinement of the information-theoretic characterization of the generalization of models obtained via SGD. This is assessed on some basic neural architectures and inspires the use of new regularizers. Overall, even though the perspective of this paper is not novel, the presented results appear to be clearer and tighter than prior instances of the same ideas. This was appreciated by most reviewers. The few clarity and organization concerns that were raised by the reviewers were adequately addressed by the authors. Overall, the paper deserves to be shared with the community.